# Investigating KDP signatures inside and below the dendritic growth layer with W-band Doppler Radar and in situ snowfall camera

Anton Kötsche<sup>1</sup>, Alexander Myagkov<sup>2</sup>, Leonie von Terzi<sup>4</sup>, Maximilian Maahn<sup>1</sup>, Veronika Ettrichrätz<sup>1</sup>, Teresa Vogl<sup>1</sup>, Alexander Ryzhkov<sup>5,6</sup>, Petar Bukovcic<sup>5,6</sup>, Davide Ori<sup>3</sup>, and Heike Kalesse-Los<sup>1</sup>

Correspondence: anton.koetsche@uni-leipzig.de

Abstract. Polarimetric radars provide variables like the specific differential phase  $(K_{DP})$  to detect fingerprints of dendritic growth in the dendritic growth layer (DGL) and secondary ice production, both critical for precipitation formation. A key challenge in interpreting radar observations is the lack of in situ validation of particle properties within the radar measurement volume. While high  $K_{DP}$  in snow is usually associated with high particle number concentrations, only few studies attributed  $K_{DP}$  to certain hydrometeor types and sizes. To address this, we combined surface in situ observations from the Video In Situ Snowfall Sensor (VISSS) with remote sensing data from a polarimetric W-band radar and an X-band radar, along with modeling approaches. Data was collected during the CORSIPP project, part of the ARM SAIL campaign (winter 2022/2023, Colorado Rocky Mountains). We found that at W-band,  $K_{DP} > 2 \,^{\circ} \, \mathrm{km}^{-1}$  can result from a broad range of particle number concentrations, between 1 and  $1001^{-1}$ . Blowing snow and increased ice collisional fragmentation in a turbulent layer enhanced observed  $K_{DP}$  values. T-matrix simulations indicated that high  $K_{DP}$  values were primarily produced by particles smaller than 0.8 mm in the DGL and 1.5 mm near the surface. Discrete dipole approximation simulations based on VISSS data suggested that dendritic aggregates larger than 2.5 mm contributed  $10\text{-}20\,\%$  to the measured W-band  $K_{DP}$  near the surface. These findings highlight the complexity of interpreting W-band  $K_{DP}$  in snowfall and emphasize the need for combined in situ observations and radar forward simulations to better understand snowfall microphysical processes.

#### 15 1 Introduction

More than 60 % of global precipitation is generated through the ice phase (Heymsfield et al., 2020); therefore, research into the microphysics of the ice and mixed phase has intensified in recent years. The dendritic growth layer (DGL), a temperature region roughly between -10 and -20 °C, plays a major role in the formation of precipitation through the ice phase. In the DGL, the difference between the saturation vapor pressure over ice and water is maximized (Wegener-Bergeron-Findeisen process (Wegener, 1911; Bergeron, 1935)), which leads to a rapid growth of ice into very anisotropic plate-like shapes (Takahashi,

<sup>&</sup>lt;sup>1</sup>Leipzig Institute for Meteorology (LIM), Leipzig University, Leipzig, Germany

<sup>&</sup>lt;sup>2</sup>Radiometer Physics GmbH, Meckenheim, Germany

<sup>&</sup>lt;sup>3</sup>Institute of Geophysics and Meteorology, University of Cologne, Cologne, Germany

<sup>&</sup>lt;sup>4</sup>Meteorological Institute, Ludwig-Maximilians-Universität in Munich, Munich, Germany

<sup>&</sup>lt;sup>5</sup>NOAA National Severe Storms Laboratory, Norman, Oklahoma, USA

<sup>&</sup>lt;sup>6</sup>Cooperative Institute for Severe and High-Impact Weather Research and Operations, University of Oklahoma, Norman, Oklahoma, USA

2014) while the number concentration is increased through secondary ice production (SIP) mechanisms (e.g. Von Terzi et al., 2022). A distinct secondary particle mode linked to the formation of new anisotropic particles may be observed at the top of the DGL as a bimodality in Doppler spectra from vertical pointing measurements (e.g. Moisseev et al., 2015). The origin of this secondary mode is still debated, theories about its formation were summarized in Von Terzi et al. (2022). Potential sources include sedimentation of ice particles, primary nucleation due to upward air motion, and some form of SIP mechanism.

Secondary ice production (SIP) has been identified as a key mechanism that enhances ice particle concentrations at temperatures above the homogeneous freezing temperature (Korolev and Leisner, 2020). SIP involves mechanisms that increase the number of ice particles without directly relying on primary ice nucleation through ice nucleating particles (INPs) (e.g., Hoose and Möhler, 2012; Kanji et al., 2017) or homogeneous freezing at very low temperatures (e.g., Sanz et al., 2013). As mountainous regions around the globe act as natural water reservoirs, storing water in the form of snow and ice, a special focus is put on microphysical processes leading to snow formation in mountain regions. Wind shear and associated turbulence were found to impact precipitation production and ice microphysics in multiple studies in recent years (e.g. Grazioli et al., 2015; Ramelli et al., 2021; Dedekind et al., 2023). Grazioli et al. (2015) proposed that splinters from SIP processes can be recirculated in the layer of shear and grow into small oblate ice particles. The area above a shear layer is supplied with supercooled liquid water (SLW) and ice crystals produced inside the turbulent updrafts. This can favor the growth of anisotropic ice crystals in this region and is suspected to cause enhanced  $Z_{DR}$  signatures (Hogan et al., 2002; Grazioli et al., 2015). Ramelli et al. (2021) investigated a turbulent layer in an inner-Alpine valley and found evidence of riming, ice collisional fragmentation and aggregation inside this layer. During another study in the Alps, Dedekind et al. (2023) recently found that SIP is best predicted by wind shear. Chellini and Kneifel (2024) found larger ice aggregates and higher concentrations of ice particles in turbulent layers of Arctic clouds, likely due to increased fragmentation. All SIP processes cause an increase in ice number concentration, and depending on the temperature range, the newly formed ice splinters can serve as embryos for the growth of new anisotropic ice particles (e.g. Grazioli et al., 2015; Luke et al., 2021).

Polarimetric radars can help to detect the fingerprints of dendritic growth in the DGL and SIP processes by transmitting and receiving polarized electromagnetic waves. The specific differential phase  $(K_{DP})$  is a polarimetric radar variable utilized across various frequency bands that is intrinsically dependent on the size and number concentration of hydrometeors. Because  $K_{DP}$  is very sensitive to high concentrations of anisotropic particles, it is a useful tool for detecting signatures of SIP in slanted radar measurements. High-frequency radars operating at smaller wavelengths, such as Ka- or W-band, are particularly effective for detecting small ice particles produced by SIP. These shorter wavelengths yield larger  $K_{DP}$  magnitudes and reduced noise compared to centimeter-wavelength radars (Bringi et al., 2001). Moreover,  $K_{DP}$  signals are easier to interpret with respect to other polarimetric variables as they are less affected by wavelength-dependent scattering effects, such as non-Rayleigh effects (Lu et al., 2015).  $K_{DP}$  can be complemented with other polarimetric variables, such as integrated differential reflectivity ( $Z_{DR}$ ), which is independent of number concentration but also sensitive to particle density and aspect ratio (Kumjian, 2013). Due to the presence of high number concentrations of anisotropic particles, areas of enhanced  $K_{DP}$  and  $Z_{DR}$  are often found as distinct layers in the DGL. These maxima are often not collocated, a discrepancy that remains under debate (e.g., Andrić et al., 2013; Moisseev et al., 2015; Oue et al., 2018; Von Terzi et al., 2022).

To explain the behavior of  $K_{DP}$  and  $Z_{DR}$  in the DGL, the different types of particles present in the upper DGL and also how they influence  $Z_{DR}$  and  $K_{DP}$  need to be considered. Schrom and Kumjian (2016) distinguished particles into two classes. The first class are isometric (I-type) particles, containing a broad variety of aggregates and ice crystals with irregular or nearly spherical shape which can be generated throughout the whole cloud depth. The second class are dendritic (D-type) particles, containing all pristine dendrites, plates and needles, generally having higher density and more anisotropic shape than I-type particles. D-type particles grow only in defined temperature regimes: from  $-20\,^{\circ}\mathrm{C}$  to  $-10\,^{\circ}\mathrm{C}$  for dendrites and hexagonal plates and between -3 °C and -8 °C for needles (Schrom and Kumjian, 2016; Griffin et al., 2018). Griffin et al. (2018) followed this particle classification of Schrom and Kumjian (2016) and developed a theory for the origin of  $Z_{DR}$  and  $K_{DP}$ signatures inside the DGL. Griffin et al. (2018) state that I-type particles can produce moderate  $Z_{DR}$  and quite significant  $K_{DP}$  if their concentration is sufficiently high, while D-type particles lead to extremely large  $Z_{DR}$  and tangible  $K_{DP}$ , if their concentration is high enough. They suggested that high  $Z_{DR}$  signatures are produced by particle populations dominated by D-type particles because of their very low (or high) aspect ratio and their high density, whereas high  $K_{DP}$  is linked to the overwhelming presence of I-type particles. Griffin et al. (2018) also made clear that I and D-type particles will mostly coexist, and that polarimetric radar variables depend on the relative contributions of the I-type and D-type ice particles in the mixture. We will revisit their hypothesis later on in our study. SIP in the DGL, although not mentioned by Griffin et al. (2018), adds an additional possibility of how a population of I-type particles with low  $Z_{DR}$  and  $K_{DP}$  signatures transitions into a mix of I and D-types with tangible  $Z_{DR}$  and  $K_{DP}$  to the pathways suggested by Griffin et al. (2018).

The primary challenge in interpreting radar variables lies in the lack of in situ validation of the properties of the particle population within the radar measurement volume. Various approaches have sought to address this, including the use of radar data in conjunction with airborne in situ measurements, as in the HALO-AC3 (Wendisch et al., 2024) and IMPACTS (Dunnavan et al., 2022) campaigns. Alternatively, surface in situ precipitation measurements have been used to validate polarimetric radar observations, as demonstrated in studies employing scanning C-band radars and a NASA Particle Video Imager (Moisseev et al., 2015), or collocated polarimetric radars and disdrometers during field campaigns such as GLACE (Grazioli et al., 2015) and TRIPEx (Dias Neto et al., 2019).

80

The unique dataset presented in our study features continuous measurements from the slanted polarimetric W-band radar (LIMRAD94, type RPG-FMCW-94-DP (Küchler et al., 2017)), situated in close proximity to the Video In Situ Snowfall Sensor (VISSS; (Maahn et al., 2024)). It enables us to directly link polarimetric radar variables to hydrometeor population properties observed by VISSS at the ground and investigate the origin of  $K_{DP}$  signatures at W-band. The data was collected within the Characterization of Orography-influenced Riming and Secondary Ice production and their effects on Precipitation rates using radar Polarimetry and Doppler spectra (CORSIPP) project, which is part of the DFG Priority Programm SPP-PROM (Trömel et al., 2021). CORSIPP was embedded in the Atmospheric Radiation Measurement (ARM) Surface Atmosphere Integrated Field Laboratory (SAIL) campaign (Feldman et al., 2023) during the very snow-rich winter season 2022/2023 in the Colorado Rocky Mountains. The data of LIMRAD94 is complemented by other collocated radars like the dual polarization Colorado State University X-band radar (CSU X-band) (Heflin et al., 2024), and the single polarization Ka-band ARM Zenith Radar

**Figure 1.** Field of view of LIMRAD94 during CORSIPP pointed at 151 $^{\circ}$  azimuth. The red arrows mark the locations of AMF2, VISSS and CSU X-band.

(KAZR, (Widener et al., 2012)). The measurement site was strongly influenced by orographic turbulence, hence we also aim to elucidate the influence of turbulence on polarimetric variables and investigate possible SIP fingerprints.

In Sect. 2, we describe the used data with a special focus on polarimetric radar variables in 2.1.1. The modeling approaches are described in Sect. 2.4. The fall streak tracking method and eddy dissipation rate retrieval are described in Sect. 2.5 and Sect. 2.6, respectively. In Sect. 3.1, we provide statistics on VISSS-derived number concentration and the ratio of the third to the second measured PSD moments  $(D_{32})$  along with collocated LIMRAD94  $K_{DP}$  measurements. In a case study of four fall streaks, we analyze  $K_{DP}$ , integrated and spectral  $Z_{DR}$ , reflectivity  $(Z_e)$ , spectral width (SW), and correlation coefficient  $(\rho_{hv})$  to identify the particles and microphysical processes contributing to the observed  $K_{DP}$  magnitudes (Sect. 3.2). Combining T-matrix modeling and CSU X-band data, we try to constrain the particle size responsible for the observed  $K_{DP}$  magnitudes (Sect. 3.3). We also aim to quantify the contribution of aggregates larger than 2.5 mm to the observed W-band  $K_{DP}$  in Sect. 3.4. The main findings are summarized in Sect. 4.

#### 2 Data and Methods

In the following sections, the instruments used during this study and their variables are described. We also provide an overview of the used modeling approaches as well as the retrievals for the fall streaks and the eddy dissipation rate.

#### 2.1 Radars

During CORSIPP, LIMRAD94 was mounted on a novel cold-temperature scanning unit, also manufactured by RPG. LIM-RAD94 was deployed at an altitude of 2905 m at the Rocky Mountain Biological Laboratory (RMBL) about 525 m away from the second ARM Mobile Facility (AMF2) in Gothic (CO) near the RMBL Ore House between November 15, 2022 and June 5, 2023. LIMRAD94 observations were collected at constant elevation of 40° and an azimuth of 151° (towards the VISSS, AMF2 and the CSU X-band radar). The temporal resolution of LIMRAD94 data is about 3 s. Further technical details on the deployment of LIMRAD94 and VISSS during CORSIPP can be found in Kalesse-Los et al. (2023). The technical details of the radar chirp program employed during CORSIPP are listed in Tab. A1. For SAIL, the ARM KAZR (Widener et al., 2012) and a scanning dual polarization X-band radar (CSU X-band) from Colorado State University (CSU) were deployed in a distance of 525 m (at 151° azimuth and 2889 m ASL) and 8018 m (at 149° azimuth and 3147 m ASL), respectively, from LIMRAD94 (also see Appendix, Table B1 and Fig. B1). From the CSU X-band radar data, RHI scans at 329° azimuth (towards the AMF2 and LIMRAD94) and plane position indicator (PPI) scans at 4° elevation were used in this study. In the appendix, Sect. B we also briefly discuss the volume matching of the CSU X-band radar and LIMRAD94 data.

#### 2.1.1 Polarimetric radar data

LIMRAD94 is capable of measuring spectrally resolved polarimetric variables like differential phase shift and differential reflectivity, among others. Polarimetric calibration of LIMRAD94 was performed at the beginning of the field campaign during a time of vertical pointing measurements as described in Myagkov et al. (2016a).

The differential phase shift  $(\Phi_{DP}, {}^{\circ})$  consists of a backscatter and a propagational part as given by:

$$\Phi_{DP}(r) = \delta(r) + 2\int_{0}^{r} K_{DP}(s) \,\mathrm{d}s \tag{1}$$

where  $\delta(^{\circ})$  is the backscatter differential phase that occurs due to non-spherical hydrometeors large enough relative to the radar wavelength such that the scattering is in the non-Rayleigh regime (Trömel et al., 2013). The specific differential phase shift  $(K_{DP}, ^{\circ} \text{ km}^{-1})$  is the range derivative of the propagational component of  $\Phi_{DP}$ . Hence, to accurately calculate  $K_{DP}, \delta$  needs to be estimated and subtracted from  $\Phi_{DP}$  first. The estimation of  $\delta$  in rain was thoroughly discussed in (e.g., Trömel et al., 2013; Myagkov et al., 2020). However, for most frozen hydrometeors,  $\delta$  was found to be negligible in the S, C, and X-bands (Balakrishnan and Zrnic, 1990; Ryzhkov et al., 2011; Trömel et al., 2013). In spectrally resolved  $\Phi_{DP}$  data,  $\delta$  can be detected by a sudden increase or decrease of  $\Phi_{DP}$  in parts of the Doppler spectrum. Consequently, also integrated  $\Phi_{DP}$  often reveals characteristic "bumps" if  $\delta$  is present (Trömel et al., 2013). During the case study presented, we checked integrated and spectral  $\Phi_{DP}$  for the above mentioned signs of  $\delta$  that may impair the interpretation of  $K_{DP}$  signatures. However, we found  $\delta$  during our case study to be negligible. For LIMRAD94 and CSU X-band, the  $K_{DP}$  was calculated from  $\Phi_{DP}$  using a convolution with low-noise Lanczos differentiators with window length of 23 gates. This method provides results that are analogue to a moving window linear regression and is implemented in the python package wradlib (Heistermann et al., 2013).

 $K_{DP}$  depends on the shape, orientation, concentration, density and the size of ice particles. High concentrations of dense particles with more anisotropic shape and uniform orientation cause high  $K_{DP}$  values. In rain,  $K_{DP}$  is proportional to the 3rd power of the particle size  $(K_{DP} \sim D^3)$ . In low density snow,  $K_{DP}$  is proportional to the 1st power of the particle size  $(K_{DP} \sim D^1)$  due to density of snow being inversely depended on the diameter  $(\rho \sim D^{-1})$ . As the degree of riming increases, the proportionality of  $K_{DP}$  shifts towards a higher power of the particle size, reflecting changes in the snow density. However, the influence of riming on  $K_{DP}$  is contradictory: while riming increases the density of particles and hence  $K_{DP}$ , riming also makes particles more spherical reducing  $K_{DP}$  (Ryzhkov and Zrnic, 2019).  $K_{DP}$  is also influenced by turbulence through its dependency on the width of the canting angle distribution  $(\sigma)$  via the orientation factor  $(F_0)$ , which is a function of  $\sigma$  (Ryzhkov et al., 2002; Ryzhkov and Zrnic, 2019). The canting angle is defined as the angle between the projection of a symmetry axis of a spheroidal particle and the vertical axis onto the polarization plane (Ryzhkov et al., 2002). If the canting angles of particles in a population differ a lot, the width of the canting angle distribution increases. The stronger the turbulence, the larger  $\sigma$ , the smaller  $F_0$  and hence the smaller  $K_{DP}$ .

The differential reflectivity ( $Z_{DR}$ , dB) is commonly defined as:

$$Z_{DR} = 10 \cdot \log_{10} \left( \frac{Ze_H}{Ze_V} \right) \tag{2}$$

where  $Ze_H$  (mm<sup>6</sup> m<sup>-3</sup>) is the equivalent radar reflectivity measured at horizontal polarization and  $Ze_V$  (mm<sup>6</sup> m<sup>-3</sup>) is the equivalent radar reflectivity measured at vertical polarization. At W-band, the largest contribution to positive  $Z_{DR}$  values (in the ice phase) is usually made by the most dense anisotropic ice, small enough to be in the Rayleigh scattering regime, usually located on the slow edge of the Doppler spectrum. Contrary to  $K_{DP}$ ,  $Z_{DR}$  is immune to the total concentration factor, so the combined analysis  $K_{DP}$  and  $Z_{DR}$  can be helpful to decouple effects of shape and concentration. Larger, strongly rimed, or aggregated hydrometeors, typically located on the fast side of the Doppler spectrum, are more spherical and exhibit low  $Z_{DR}$ . Since  $Z_{DR}$  is reflectivity-weighted, its magnitude is often reduced by the influence of these larger, spherical particles (e.g. Oue et al., 2018). To use  $Z_{DR}$  without suffering from this masking effect, we also calculate the maximum spectral  $Z_{DR}$  value  $(sZ_{DR_{max}})$  for each Doppler spectrum. As demonstrated in Von Terzi et al. (2022), this parameter is efficient for detection of small non-spherical particles masked by presence of large, more spherical particles like aggregates. Similar to  $K_{DP}$ ,  $Z_{DR}$ is depend on  $\sigma$  (Hubbert and Bringi, 2003; Ryzhkov and Zrnic, 2019) and hence also decreased by turbulence.  $sZ_{DR_{max}}$  is furthermore decreased by turbulence through the broadening of the Doppler spectrum. Imagine that only one type of particle in a radar volume, moving with a certain terminal fall velocity, produces  $Z_{DR}$  larger than zero.  $sZ_{DR_{max}}$  is high because the positive  $Z_{DR}$  is measured at one certain Doppler velocity bin. If the fall velocity is now determined by turbulent air motion rather than the particles inherent terminal fall velocity, the particles producing the positive  $Z_{DR}$  are "distributed" over a larger range of Doppler velocities and the  $Z_{DR}$  at one certain Doppler velocity bin decreases. It is therefore essential to consider turbulence before interpreting any  $K_{DP}$  or  $Z_{DR}$  signatures.

LIMRAD94 was measuring at 40° elevation, which will cause  $Z_{DR}$  and also  $K_{DP}$  to be smaller than at lower elevation angles (e.g., Myagkov et al., 2016b). This should be taken into account when comparing the bulk values with other studies using  $Z_{DR}$  and  $K_{DP}$  but at different elevation angles.

#### 170 2.2 VISSS

The first generation VISSS was installed next to the AMF2 facility, in the line of sight of LIMRAD94 with a horizontal distance to LIMRAD94 of approximately 550 m (Fig. 1). VISSS1 has a pixel resolution of 58.832  $\mu$ m px<sup>-1</sup>, a frame rate of 140 Hz, and an observation volume of  $(wxdxh) = 75.2 \times 75.2 \times 60.1 \text{ mm}^3$  (Maahn et al., 2024). This yields an observational volume of 0.000339 m<sup>3</sup>. VISSS level2match variables (see Fig. 2 in Maahn et al. (2024)) used in this study include particle size distributions (PSD),  $D_{32}$ , complexity (c) and total number concentration  $(N_{tot})$  and slope of the PSD  $(\Lambda)$ . Following (Maahn et al., 2024), the complexity c is derived from the ratio of the particle perimeter p to the perimeter of a circle with the same area A:

$$c = \frac{p}{2\sqrt{\pi A}}. (3)$$

The slope parameter of the PSD ( $\Lambda$ ) is derived using the definition of Maahn et al. (2015):

$$180 \quad \Lambda = N_0/N_{tot} \tag{4}$$

where  $N_0$  is the intercept parameter of the exponantial PSD.

 $D_{32}$  is the ratio of the third to the second measured PSD moment. Assuming that particle mass is proportional to the particle maximum dimension squared,  $D_{32}$  is a proxy for the mass-weighted mean diameter of the particle population (Maahn et al., 2015). Complexity is derived from the ratio of the particle perimeter to the perimeter of a circle with the same area (Maahn et al., 2024). The total number concentration reaches values of over  $1001^{-1}$  in this case study, which may seem unusual at first when comparing to other literature. VISSS is however more sensitive to small particles than laser disdrometers as shown by Maahn et al. (2024). This is likely the reason for the observed particle concentrations. All diameters used refer to the maximum diameter ( $D_{max}$ ), unless otherwise specified.

#### 2.3 Additional products

During our study, we used additional data acquired at the AMF2 such as 2 m temperature, pressure, relative humidity, and wind speed as well as ARM 3-channel microwave radiometer liquid water path (*LWP*) and integrated water vapor (*IWV*); a detailed list can be found in the code and data availability section. Vertical temperature profiles on site are provided by the ARM interpolated sonde and gridded sonde VAP (Fairless et al., 2021). The sounding data, collected on site through two radiosonde launches per day (11 and 23 UTC) is transformed into continuous daily files with 1-minute time resolution and combined with ARM 3-channel microwave radiometer temperature data.

#### 2.4 Modeling of radar observables

To interpret radar observations and constrain quantitative estimates of the microphysical properties of ice particles, scattering simulations are employed.

#### 2.4.1 T-matrix

The first approach approximates the shape of a particle as a spheroid and simulates its backscattering and forward scattering properties using the T-matrix scattering model (Mishchenko, 2000), hereafter referred to as the T-matrix approach. For a given wavelength, this method requires only three parameters: the equivolumetric diameter, the axis ratio, and the refractive index. The refractive index, which depends on the fraction of ice and air (Oguchi, 1983), can be adjusted to model ice particles with varying ice densities (Myagkov et al., 2016b). For simplicity, we assume the horizontal alignment of ice particles, albeit acknowledge that a broader distribution of canting angles generally leads to less pronounced polarimetric signatures. The T-matrix approach is computationally efficient; however, it is most applicable when the particles are smaller than the radar wavelength. Large ice particles, which tend to have highly irregular shapes, are not well-represented by the spheroidal approximation, particularly for short radar wavelengths. This limitation makes the T-matrix approach less suitable for modeling the scattering of rimed particles and aggregates at millimeter wavelengths (Leinonen et al., 2012; Kneifel et al., 2015).

#### 210 2.4.2 DDA

For simulating the scattering of large ice particles at millimeter wavelengths, the Discrete Dipole Approximation (DDA) is often employed. This method allows for an accurate approximation of the scattering properties of individual particles with arbitrary shapes. However, DDA is computationally intensive and requires detailed specifications of the particle geometry which is generally unknown. Here, we randomly selected 105 unrimed dendritic aggregates from the snowScatt database (Ori et al., 2021), spanning maximum dimensions of  $100 \, \mu m$  to  $4 \, cm$ . As the falling behavior of snowflakes is determined by the complex interactions between aerodynamical and gravitational forces and is further complicated by e.g., turbulence, we assume that the snowflakes fall with their maximum extension oriented horizontally. The DDA for the dendritic aggregates was calculated using *adda* (Yurkin and Hoekstra, 2011) at 94 GHz and  $40^{\circ}$  elevation, and averaged over 16 azimuth orientations.

#### 2.4.3 Rationale for the applicability of scattering models

The primary aim of employing scattering models in this study is to identify the types of ice particles that contribute to the observed  $K_{DP}$  signatures, rather than to establish a retrieval method. Since the T-matrix approximation uses spheroids, a clear relation between the  $K_{DP}/Z_e$  ratio and the size and aspect ratio of the spheroid for a fixed apparent ice density is possible. While this approach is simplified, it yields sufficiently accurate results for preliminary conclusions, particularly when the particles are smaller than the radar wavelength, where the accurate mass distribution inside of the simulated particle is not as important as when particle size and wavelength become comparable. As for DDA we use more detailed shapes, fixing the effective density is not possible and such a clear relationship can not be found. Further, we do not have precise information

about the size or shape of the observed particles. We therefore applied the spheroidal approximation and T-matrix model at X-band, the lowest frequency available in our campaign with a collocated measurement volume, to assess the sensitivity of  $K_{DP}$  and  $Z_e$  to particle size and aspect ratio (Sect. 3.3). This analysis also aids in determining whether  $K_{DP}$  signatures are predominantly generated by larger particles. Conversely, the T-matrix method produces larger errors at W-band frequencies, where the shorter wavelength means non-Rayleigh scattering effects and the accurate description of the mass distribution in the particle become relevant (e.g. Ori et al., 2021, and references therein). This is especially true for particles with a low density, such as aggregates. Describing an aggregate with a soft spheroid is problematic, as the effective density in the spheroid needs to be small to accurately match the size and mass of the aggregate. This causes the (polarimetric) scattering properties of aggregates to be underestimated by soft spheroid methods such as the Tmatrix (e.g. Ori et al., 2021). Consequently, to estimate  $K_{DP}$  and  $Z_e$  at W-band for the particles detected with the VISSS, particularly concentrating on dendrite aggregates, we opt to use DDA computations (Sect. 3.4).

#### 2.5 Fall streak tracking

Hydrometeor populations are influenced by horizontal and vertical winds as they precipitate to the surface and are hence exposed to advection. To estimate their microphysical evolution and to link ground observations with radar measurements aloft, these hydrometeor populations need to be observed along slanted fall streaks in vertical radar measurements. The analysis of the fall streaks is mostly achieved by reconstructing the advection using the present wind measurements (Marshall, 1953; Kalesse et al., 2016; Pfitzenmaier et al., 2017). As no sufficiently high resolution wind data is available for our case study, we decided to analyze fall streaks with the simple approach of following maximum  $Z_e$  values backwards from surface to cloud top. Specifically, the  $Z_e$  at the start point is compared to the median  $Z_e$  of the next three higher range gates. The time difference between two neighboring range gates is not allowed to exceed  $\frac{2\Delta h}{v_t}$  where  $\Delta h$  is the height difference in m between the range gates and  $v_t$  is the mean Doppler velocity (MDV) measured by KAZR. The maximum  $Z_e$  is chosen as the next point along the fall streak, and the process is repeated. This iterative process determines the suspected course of the fall streak in time and height. We acknowledge that horizontal homogeneity has to be assumed, especially for slanted measurements. Because the main wind direction in our case study is westerly, but the radar is pointing south-east it is obvious that we will never observe the exact same particles along their way to the ground.

#### 2.6 Turbulence eddy dissipation rate retrieval (EDR)

Turbulence plays a critical role in understanding the evolution of hydrometeor populations and the behavior of polarimetric radar variables. The turbulence eddy dissipation rate (EDR) is used as a proxy for turbulence. EDR is the rate at which turbulence kinetic energy dissipates in the atmosphere (Foken and Mauder, 2024). High EDR values indicate a more turbulent environment. We calculated EDR from the KAZR MDV for five minute time intervals, using the approach described in (Vogl et al., 2024). To mask turbulence, we applied an empirical threshold of  $0.001 \, \mathrm{m}^2 \mathrm{s}^{-3}$  similar to the one found by Vogl et al. (2022). Note that the radar beams of KAZR and LIMRAD94 during CORSIPP were not aligned, so the measurement

Figure 2. Scatterplot of LIMRAD94  $K_{DP}$  vs. VISSS  $N_{tot}$  for DJF 2022/23. The x-scale is logarithmic. LIMRAD94  $K_{DP}$  was spatially averaged between 100 and 500 m above ground and temporally averaged to fit the VISSS time resolution of one minute. Colors show  $D_{32}$  of the respective PSD. A total of 14773 data points (14687 minutes) were used for the plot.

volume from which EDR is retrieved has a horizontal displacement from the LIMRAD94 measurement volume. This distance increases with increasing height.

#### 3 Results and Discussion

First, we show a statistical comparisons of VISSS  $N_{tot}$  and LIMRAD94  $K_{DP}$  (Sect. 3.1). A case study with four  $K_{DP}$  fall streaks is shown in (Sect. 3.2). For this case study, we present a combination of T-matrix  $K_{DP}$  modeling and collocated CSU X-band observations in (Sect. 3.3) and  $K_{DP}$  simulations using DDA based on VISSS data (Sect. 3.4).

#### 265 3.1 $K_{DP}$ vs VISSS statistics

One advantage of the CORSIPP dataset is the close proximity of in situ observations by VISSS and detailed polarimetric W-band measurements of LIMRAD94. To understand how  $K_{DP}$  is related to properties of the PSD, we compare  $K_{DP}$  measurements of LIMRAD94 to VISSS surface measurements of  $N_{tot}$  and  $D_{32}$  for the period December 1, 2022 to February 28, 2023. LIMRAD94  $K_{DP}$  was spatially averaged (median) between 100 and 500 m above ground (52 range gates) to reduce effects of the measurement error, for the horizontal displacement of the two measurement volumes and for inhomogeneities in the observed precipitation. LIMRAD94  $K_{DP}$  was also temporally averaged (median) to match the VISSS time resolution of one minute. To compensate for the vertical displacement of the measurement volumes of VISSS and LIMRAD94 and the resulting time lag, LIMRAD94 data was shifted backwards in time as precipitation would be first visible in LIMRAD94 measurements and later in the VISSS. Given that the center of the averaged height is 300 m and assuming a terminal fall velocity of 1 m s<sup>-1</sup>, we chose five minutes as value for the time shift. We also tested time shifts of 3 minutes and 10 minutes; none of them significantly changed the overall data distribution (not shown). Negative  $K_{DP}$  values and range gates with SNR 

**Figure 3.** Meteorological observations during the case study on December 6, 2022. The vertical dashed magenta line in all plots indicates the passing of a cold front. a): Horizontal wind speed and direction. Horizontal grey dashed lines represent air temperature in  $^{\circ}$ C from the interpolated radiosonde product. The analyzed fall streaks are displayed as black lines with their respective labels. The black solid horizontal line denotes the height of Gothic Mountain. b) AMF2-based meteorological variables at 2 m: pressure, wind speed and direction as colorcode. c): AMF2-based 2 m temperature and relative humidity. d): AMF2-based MWR LWP and IWV.

characterized by enhanced  $K_{DP}$ . Despite exhibiting similar  $K_{DP}$  magnitudes, VISSS revealed varying particle population properties within these fall streaks.

#### 3.2.1 Synoptic discussion

December 6, 2022 featured a complex synoptic situation with a stationary front over the Colorado Rocky Mountains and a surface low developing along this front over eastern central Colorado, slowly moving southward during the day. The measurement site was influenced by the development of this surface low. An overview of surface meteorological variables and the vertical wind profile during the case study is given in Fig. 3. Just before 15 UTC we observed a sudden change in 2 m wind direction from south-west to north winds, an increase in 2 m wind speed from under 2 m s<sup>-1</sup> to over 5 m s<sup>-1</sup> (Fig. 3b) and a decrease in 2 m temperature and humidity (Fig. 3c). After 15 UTC, we saw a steady increase in surface pressure (Fig. 3b). The evolution of the surface meteorological variables in Gothic points to a cold front passage just before 15 UTC (see magenta lines in Fig. 3).

The wind throughout the lower troposphere up to 3000 m above LIMRAD94 was from the south-west before the cold front with a slight veer and an increase in speed toward higher altitudes, indicating the advection of warm air (Fig. 3a). With the passage of the cold front, the wind turned to westerly and later north-westerly directions up to 1500 m above LIMRAD94. Aloft, the winds backed to south-westerly direction again, indicating cold air advection. LWP and IWV decreased with the passage of the cold front (Fig. 3d), indicating that a dryer air mass being advected. The nature of the precipitation changed during the cold front passage; from low  $N_{tot}$ , low  $\Lambda$  and high  $D_{32}$  precipitation dominated by aggregates and comparably low number of small particles before the cold front to a high  $N_{tot}$ , high  $\Lambda$  and low  $D_{32}$  precipitation with very mixed particle types (see Fig. 4e). Therefore, we especially focus on this time period between 14:30 and 16:30 UTC in our case study.

#### 315 3.2.2 The turbulence layer

Radar variables observed by LIMRAD94 as well as VISSS  $N_{tot}$ ,  $\Lambda$  and  $D_{32}$  throughout the case study are shown in Fig. 4. Due to the orography surrounding the measurement site, we observed a distinct layer of enhanced turbulence, mostly at and below  $1000 \,\mathrm{m}$  above ground level (AGL). This is visualized by the EDR-masked data (white where EDR exceeds  $0.001 \,\mathrm{m}^2 \,\mathrm{s}^{-3}$ ) in Fig. 4c).

The main wind direction above 1000 m AGL was south-west before the cold front and west to north-west after the cold front passage (Fig. 3a). The inner valley flow was weak south-westerly before the cold front and moderate northerly after the cold front (Fig. 3b). Hence, the wind shear was enhanced along the interface between the blocked valley and the cross-barrier flow aloft, which would lead to shear-induced turbulence as explained in Ramelli et al. (2021). The main blocking feature west of Gothic is Gothic Mountain (3850 m). The height of the summit of Gothic Mountain above the location of LIMRAD94 (roughly 1000 m AGL) corresponds well to the top of the observed turbulence layer.

Fig. 5 shows the MDV of CSU X-band from a PPI (a) and a RHI (b) scan at around 15:48 UTC and 15:46 UTC, respectively. During that time, *EDR* indicates enhanced turbulence between 500 and 1200 m AGL (Fig. 4c). The MDV measured during the PPI scan can be approximated as the radial component of the horizontal wind. Negative values (blue) in Fig. 5 correspond to motion towards the radar, positive values (yellow-red) to motion away from the radar. The PPI is partly blocked by mountains, however the westerly wind is apparent by blue colors west of CSU X-band and orange-red colors to the east. Where MDV is close to zero, the wind is perpendicular to the radar beam. Downstream of Gothic Mountain, there is a small corridor where the MDV values turn positive in sharp contrast to the surrounding negative values. This implies that downstream of Gothic Mountain, the wind turns to more southerly directions. The effect diminishes quickly further downstream. In the RHI scan (Fig. 5b), the southerly wind component is visible by the area of more green colors at 6000-7000 m range from CSU X-band and between 400-1000 m above LIMRAD94. This area of lee-induced flow disturbance (termed ALIFD) could be partly caused by a counterflow, indicative of a blocked low-level flow, as, for example, described in Ramelli et al. (2021). Along the edges of the ALIFD, wind shear is enhanced which could lead to the observed turbulence. The beam of LIMRAD94 is shown in Fig. 5b) as a red cone, and it is apparent that the measurements of LIMRAD94 are impacted by the turbulence in and around the ALIFD.

Figure 4. LIMRAD94 time-height plots during the case study on December 6, 2022. Analyzed fall streaks shown as black lines with labels. In every panel except d), data is only colored where LIMRAD94 KPD >  $2^{\circ}$  km<sup>-1</sup>. The remainder of the  $Z_e$  and  $sZ_{DR_{max}}$  /  $Z_{DR}$  observations where  $K_{DP}$  was below this threshold are displayed in black and white. Grey lines are isotherms. a)  $Z_e$ . b)  $Z_{DR}$ . c)  $sZ_{DR_{max}}$  with areas masked where EDR exceeds the turbulence threshold of 0.001 m<sup>2</sup> s<sup>-3</sup>. d)  $K_{DP}$ . e) Minutely VISSS  $N_{tot}$  on the y-axis and  $\Lambda$  (black line) plotted against time on the x-axis. Colors are the respective  $D_{32}$  values.

Figure 5. a) Mean Doppler velocity (approx. radial component of horizontal wind) of CSU X-band PPI scan at  $4^{\circ}$  elevation angle with location of Gothic Mountain (white circle) and direction of RHI scan at azimuth  $329^{\circ}$  (black line) at 15:48:32 UTC on Dec 6, 2022. Grey dashed circles are placed at every 1000 m range from CSU X-band. Labels are the respective heights above LIMRAD94 of the CSU X-band radar beam. b) Mean Doppler velocity of CSU X-band RHI scan (azimuth  $329^{\circ}$ ) at 15:46:21 UTC on Dec 6, 2022 with LIMRAD94 beam direction (red). Grey shading shows the elevation profile.

#### 340 3.2.3 Evolution of polarimetric variables along fall streaks (FS)

In the following sections, the development of W-band polarimetric variables along the four fall streaks of enhanced  $K_{DP}$  before, during, and after the cold front are analyzed in detail. VISSS images of hydrometeors during the fall streaks and the respective PSD are shown in Fig. 6 and Fig. 7, respectively. The polarimetric radar observations during the four selected fall streaks (FS1-4) are shown in Fig. 8. All subplots in Fig. 8 have the same structure, showing profiles of  $K_{DP}$ ,  $Z_e$ ,  $sZ_{DR_{max}}$ ,  $Z_{DR}$ , spectral width (SW) and correlation coefficient ( $\rho_{hv}$ ) as a function of the temperature. Fall streaks were derived as explained in Section 2.5. A grey line was added for every height / temperature where EDR exceeded 0.001 m<sup>2</sup>s<sup>-3</sup>. SW of LIMRAD94 was added to the plots as an additional proxy for turbulence (as for example done in Majewski et al., 2023). For each of the analyzed fall streaks, we will try to assess which particle types and microphysical growth processes lead to the observed  $K_{DP}$  signatures and how they are influenced by turbulence.

#### 350 Before the cold front (FS1)

The first  $K_{DP}$  fall streak (FS1) we analyze appears around 14:15 UTC, originating in the DGL at around 2300 m AGL and temperatures between -20 and -15 °C (Fig. 4d). It reaches the ground well before the cold front at around 14:30 UTC. Before the cold front passage, the dominant particle type is aggregates, smaller heavily rimed or graupel-like particles were also observed by the VISSS (see Fig. 6a). Before 14:45 UTC,  $N_{tot}$  is always below  $101^{-1}$  and  $D_{32}$  is around 4-5 mm (Fig. 4e),

**Figure 6.** Particles observed on the ground by VISSS during the four fall streaks. For each fall streak, 3 minutes of data are displayed. Particles were divided into size bins and a fixed number of particles is displayed for each bin (if the bin contains a sufficient number of particles). The number of displayed particles is decreasing with increasing particle size.

indicating that the particle population at the surface contains larger particles (see also Fig. 7). A low  $\Lambda$  (Fig. 4e) of around  $0.5 \,\mathrm{mm}^{-1}$  further suggests a PSD with a reduced number of smaller particles, which is also visible in Fig. 7).

The polarimetric variables along FS1 are shown in Fig. 8a). At temperatures colder than  $-20\,^{\circ}$ C,  $K_{DP}$  is close to zero and  $sZ_{DR_{max}}$  rather low with values around 0.25 dB.  $Z_{DR}$  is slightly negative with -0.2 dB. The negative  $Z_{DR}$  observed above the DGL is most likely caused by differential attenuation within the DGL. Differential attenuation at W-band by ice is either related to large particle number concentrations or high mass densities or a combination of both (Liao et al., 2008).  $\rho_{hv}$  in this upper part of FS1 is high with values above 0.995. This combination of  $Z_{DR}$ ,  $K_{DP}$  and  $\rho_{hv}$  suggests that the particle population is rather uniform and consists either of nearly spherical particles, particles with a low effective refractive index or a combination of the two (e.g. Andrić et al., 2013).

As these particles fall into the DGL, they experience depositional growth through enhanced supersaturation over ice.  $sZ_{DR_{max}}$  and  $Z_{DR}$  increase because depositional growth in the DGL promotes anisotropic shape enhancement (e.g. Andrić et al., 2013; Takahashi, 2014). The sharp decrease in  $\rho_{hv}$  together with the difference in  $sZ_{DR_{max}}$  and  $Z_{DR}$  could be linked to a secondary particle mode consisting of freshly formed, anisotropic particles is often observed in the DGL (described in Sect.

Figure 7. VISSS PSDs (one minute averages) observed during the times when the four selected fall streaks reached the surface.

1).  $sZ_{DR_{max}}$  starts to increase at around  $-20\,^{\circ}\mathrm{C}$  to values of around  $0.75\,\mathrm{dB}$  (center panel in Fig. 8a). The 99th percentile of  $sZ_{DR_{max}}$  during the 2.5 hours of our case study is 1.32 dB, so 0.75 dB is still comparably small. The comparably low  $sZ_{DR_{max}}$  in the DGL during FS1 indicates either more spherical, less dense or smaller anisotropic particles. Collocated with the maximum  $sZ_{DR_{max}}$  and  $Z_{DR}$ , we see a sharp increase in  $K_{DP}$  (Fig. 8a) to over  $6^{\circ}$  km<sup>-1</sup>). Following the hypothesis of Griffin et al. (2018) described in the Introduction, the peaks of low  $Z_{DR}$  and high  $K_{DP}$  in FS1 between -20 and  $-15\,^{\circ}\mathrm{C}$  could be caused by a high concentration of I-type particles. If I-type particles enter the DGL, they will experience rapid depositional growth because of the enhanced supersaturation over ice. Larger particles will grow slower and mostly maintain their shape, while smaller I-types will grow faster and could serve as embryos of rapid growing D-type particles (e.g. Sheridan et al., 2009; Griffin et al., 2018). The existing water vapor would be distributed over a large number of particles, which, as we speculate, would result in overall lower supersaturation and cause the newly formed D-type particles of larger quantities to grow into overall lower diameters and lower density. So while  $Z_{DR}$  could be masked by more spherical I-type particles,  $sZ_{DR_{max}}$  in FS1 could be low because of D-type particles with an overall smaller diameter. At the same time, these particles are more numerous, which enhances  $K_{DP}$  and causes a collocation of the maximum  $sZ_{DR_{max}}$  and  $K_{DP}$ . This theory would imply that the closer the maxima in  $K_{DP}$  and  $sZ_{DR_{max}}$  are in height, the higher the overall number concentration of particles in the upper DGL. Another potential cause of the observed radar signatures could be additional SIP at the upper part of the DGL as e.g. suggested in Andrić et al. (2013) and Von Terzi et al. (2022). A large number of I-type particles with a wider size distribution would also enhance the likelihood of ice collisional fragmentation, providing even more embryos for the growth of D-type particles. A

Figure 8. Profiles of polarimetric variables along the FS1 (a), FS2 (b), FS3 (c) and FS4 (d). Profiles of  $K_{DP}$  (red),  $Z_e$  (black),  $sZ_{DR_{max}}$  (blue), Spectral Width (orange) and  $\rho_{hv}$  (green) are plotted against the temperature. All polarimetric variables were averaged (median) for +/-2 min around the fall streak track (solid lines). Colored shaded areas mark 25/75th percentile values of the respective variable. Grey shaded areas mean that the EDR exceeded the turbulence threshold of 0.001 m<sup>2</sup> s<sup>-3</sup>.

second, smaller peak of  $K_{DP}$  at around  $-15\,^{\circ}\mathrm{C}$  in FS1 could be caused by additional dendritic growth at this temperature. A similar structure of  $K_{DP}$  in the DGL was for example shown in simulations by Andrić et al. (2013), caused by dendritic growth at this temperature. At temperatures warmer than -15  $^{\circ}\mathrm{C}$ , we see an overall decrease in  $K_{DP}$  towards the ground with a collocated moderate increase of  $Z_e$ , a signal commonly attributed to the aggregation of anisotropic particles into less dense, more spherical shapes and the reduction of  $N_{tot}$  (e.g. Andrić et al., 2013; Trömel et al., 2019; Von Terzi et al., 2022; Hu et al., 2024).

Figure 9. Time-height plots of KAZR MDV (panel a and c) and KAZR  $Z_e$  (panel b and d) of the December 6, 2022 case study. The red contour in panel b and d marks the  $0 \, \mathrm{m \, s} - 1$  isotach. Dashed grey lines in panel b and d mark the isotherms (in  $^{\circ}$  C) with respective labels. The analyzed fall streaks retrieved from LIMRAD94 measurements are displayed as black lines with their respective labels. Note that these fall streak paths do not exactly match the KAZR  $Z_e$  because of the horizontal displacement of the measurement volumes.

The summit height of Gothic Mountain is roughly located around the  $-10\,^{\circ}\mathrm{C}$  isotherm. Turbulence is found at this height (Fig. 8a). Just above the turbulent layer, we see an increase in  $sZ_{DR_{max}}$  together with a small peak in  $K_{DP}$ . A similar signature of enhanced  $Z_{DR}$  above a turbulent layer was found by (Hogan et al., 2002; Grazioli et al., 2015). They explained that SLW and ice splinters are transported upward by updrafts and grow into anisotropic particles. In our case, the splinters would grow into D-type particles given temperatures between -10 and  $-15\,^{\circ}\mathrm{C}$  which might explain the enhanced  $sZ_{DR_{max}}$  and  $K_{DP}$ . Below the turbulent layer, we find that  $sZ_{DR_{max}}$  and also  $K_{DP}$  increase slightly again while  $Z_{DR}$  remains almost constant. LWP up to  $200\,\mathrm{g\,m^{-2}}$  (see Fig. 3d) during FS1 enables riming, and graupel is detected by VISSS (see Fig. 6 FS1). The enhanced  $sZ_{DR_{max}}$  and  $K_{DP}$  could be attributed to graupel-snowflake collisions, which are known to produce anisotropic ice fragments with sizes up to 2 mm (Grzegorczyk et al., 2023). In the first stage of riming, only the density of particles increases without changes in density of shape (Erfani and Mitchell, 2017), which might also cause the observed increase in  $sZ_{DR_{max}}$  and  $K_{DP}$  below the turbulent layer. The PSD observed by VISSS during the time FS1 reaches the surface shows a comparably low

number of small particles (Fig. 7) together with the presence of larger aggregates (see Fig. 6 FS1). Aggregation likely causes a rapid depletion of small particles near the surface, thereby reducing  $N_{tot}$ . The low  $\Lambda$  observed in Fig. 4 provides additional evidence that small particles are depleted in the PSD.

#### 405 During the cold front (FS2)

With the passage of the cold front at the surface, we see a distinct increase in  $N_{tot}$  in the VISSS measurements from less than  $101^{-1}$  to around  $1001^{-1}$  (Fig. 4e). At the same time,  $D_{32}$  (colors in Fig. 4e) decreases to around 2 mm and  $\Lambda$  increases to nearly 2 mm<sup>-1</sup>, indicating smaller particles are dominant in the measurement volume. Compared to FS1 before the cold front, the VISSS PSD in FS2 (Fig. 7), shows an overall increase/decrease in particles smaller/larger than 2 mm, respectively. The sudden increase in the number of small particles could be caused by blowing snow, which is supported by multiple observations. As the hours before the cold front featured conditions with weak 2 m winds and large aggregates, loose snow on trees or the ground was likely lifted along the convergence of the surface cold front and behind it by the suddenly increased northerly wind. KAZR data during the case study is shown in Fig. 9. The lift associated with the cold front can be seen as positive KAZR MDV (i.e., upward motion) in Fig. 9b. The strongest upward motion was observed at around 14:52 UTC in the lowest 500 m along the surface convergence. The low-level uplift provided by the cold front supports the theory that blowing snow was lifted and kept elevated in the vicinity of the surface convergence. In the presented case, blowing snow could also be detectable at higher altitudes because it is lifted off the higher valley slopes and advected over the lower valley floor. Figure . 9a shows the KAZR  $Z_e$  together with the 0 m s<sup>-1</sup> isotach (red line). Between 14:50 and 14:55 UTC and roughly between 250 and 900 m AGL, there is an area of enhanced  $Z_e$  with no connection to a FS, supporting the idea of  $Z_e$ -enhancement caused by blowing snow.

Between 14:50 and 14:55 UTC, we also see a small area of enhanced LIMRAD94  $K_{DP}$  in the lowest 500 m (Fig. 3b). Figure . 8b shows a possible fall streak track during the time of the cold front passage. FS2 displays a sudden change in the polarimetric variables and hence likely the particle mode when entering the pocket of cool and dryer air between -8 and -5 °C (Fig. 8b). The boundary between the two air masses is marked by an increase in turbulence (Fig. 8b). Below the top of the turbulent layer,  $sZ_{DR_{max}}$  and  $K_{DP}$  decrease sharply which could however be due to a turbulence masking effect. Within the turbulent layer,  $K_{DP}$  then increases again downwards to values of up to 3 ° km<sup>-1</sup> at the lower edge of the turbulent layer. Assuming that  $K_{DP}$  is still masked by turbulence, the true  $K_{DP}$  values without turbulence would be much higher.  $Z_e$  also increases by about 5 dBZ within the turbulence layer, its peak value is collocated with the  $K_{DP}$  peak. We conclude that this  $K_{DP}$  signature is likely caused by blowing snow.

At the top of the turbulent layer, we also see an increase in  $sZ_{DR_{max}}$  together with a small peak in  $K_{DP}$ , similar to FS1. The explanation could again be small ice splinters being transported upward by updrafts (Hogan et al., 2002; Grazioli et al., 2015). Ice splinters could originate either from SIP processes inside the turbulent layer or from lifted blowing snow. At temperatures between -5 and -8 °C the splinters can grow into columns or needles (Bailey and Hallett, 2009). Indeed, some needles were observed by the VISSS for FS2 (Fig. 6b).

#### After the cold front (FS3, FS4)

FS3 forms between 14:45 and 15:00 UTC at around 2000 m and reaches the ground around 15:10 (FS3). FS4 forms between 15:25 UTC and 15:35 UTC at 1500-2000 m and reaches the surface just after 15:45 UTC (FS4). The surface relative humidity (RH) decreases noticeably before each fall streak reaches the ground (Fig 3c) and increases when the FS reaches the ground. At the same time, surface  $N_{tot}$  increases (Fig 4e). This could point to drier air that is advected behind the cold front, which is moistened again by the sublimation of precipitating snow.

FS3 originates in the upper dendritic growth layer (DGL) at around 2000 m and temperatures between -15 and -20 °C. The maximum  $K_{DP}$  values in the DGL reach 5 ° km<sup>-1</sup> (Fig. 8c), which is slightly lower than in FS1. The peaks of  $sZ_{DR_{max}}$  and  $Z_{DR}$  have values of around 1 dB and 0.5 dB, respectively which is higher than in FS1. Comparing  $Z_e$  at -20 °C in FS1 and FS3, we find  $Z_e$  to be about 6 dBZ in FS1 and 2 dBZ in FS3. Assuming that the size of I-type particles is similar in both FS, this would point to a lower concentration of I-type particles sedimenting in the DGL in FS3 and thus less I-types serving as embryos for D-type growth. This in turn would lead to larger D-type particles (since the existing water vapor is distributed over fewer particles) with larger  $sZ_{DR_{max}}$  (and higher  $Z_{DR}$  due to less masking) and still in a sufficiently high number concentration to produce high  $K_{DP}$ .

In FS3, the increase in  $sZ_{DR_{max}}$  and  $K_{DP}$  coincides with a sharp decrease in  $\rho_{hv}$  (Fig. 8c) which quickly recovers near  $-15\,^{\circ}\text{C}$ , while  $K_{DP}$  experiences a sudden drop of approximately  $2\,^{\circ}\text{km}^{-1}$ . This signature could indicate rapid aggregation or riming, processes typically associated with an increase in  $Z_e$ . However, during the  $K_{DP}$  decrease,  $Z_e$  also shows a slight decline. This behavior may reflect the limitations of interpreting a three-dimensional system with two-dimensional data. Between  $-15\,^{\circ}\text{C}$  and  $-12\,^{\circ}\text{C}$ ,  $Z_e$  increases strongly, indicating a possible increase in particle diameter which might point to aggregation. In the same T-range,  $K_{DP}$  is nearly constant with high values of around  $3\,^{\circ}\text{km}^{-1}$ . This points to the formation of new anisotropic particles in parallel to the aggregation process, which continues to enhance the number concentration and hence keeps  $K_{DP}$  constant. Nevertheless, it is not ruled out that aggregates contribute to the magnitude of the  $K_{DP}$  in this part of FS3.

The whole area of enhanced  $K_{DP}$  around the black line marking FS3 between 14:40 and 15 UTC reveals another interesting feature: inside the area of high  $K_{DP}$  values, a horizontal gradient in  $Z_e$ ,  $Z_{DR}$  and  $sZ_{DR_{max}}$  is visible (Fig. 4a-d).  $Z_{DR}$  and  $sZ_{DR_{max}}$  are especially low on the left side of black line marking FS3 while  $K_{DP}$  exceeds  $4^{\circ}$  km<sup>-1</sup>. Following the hypothesis of Griffin et al. (2018), this might point to I-type particles seeding in the DGL in parts of the fall streak and dominating the particle population.  $Z_{DR}$  might additionally be masked by more spherical, low density aggregates. The presence of these larger aggregates could also explain the higher  $Z_e$  on the left side of FS3. On the right side of FS3, higher  $sZ_{DR_{max}}$  and  $z_{DR}$  might be caused by the presence of dense, anisotropic D-type particles. The lower  $z_e$  and less masked  $z_{DR}$  might be related to the absence of large aggregates.

Below the turbulent layer in FS3,  $K_{DP}$  decreases, but still has values of 2-2.5 ° km<sup>-1</sup> until the fall streak reaches the surface, while  $Z_e$  increases slightly towards the surface (Fig. 8c). Where  $K_{DP}$  decreases between -10 and -7 °C, we also see an increase in  $\rho_{hv}$  which might point to enhanced aggregation due to the turbulent layer as for example observed by Chellini

and Kneifel (2024). The observation of graupel, dendrites, and many dendrite fragments by VISSS (see Fig. 6c) points to frequent graupel-snowflake collisions. These collisions were found to produce up to 500 fragments per collision Grzegorczyk et al. (2023). The fragment size distributions of such collisions revealed a mode at around 0.4 mm in their study. In the PSD measured by VISSS (Fig. 6 FS3), particles with a  $D_{max}$  of less than 0.5 mm were the most frequently observed. A portion of these particles might be fragments of graupel-snowflake collisions, which could also explain the very high  $N_{tot}$  of around  $1101^{-1}$  at the surface during FS3.

In FS4 (see Fig. 8d), we find a very prominent peak in  $sZ_{DR_{max}}$  of 1.25 dB between -17 and -13 °C. Contrary to the previous fall streaks, this peak is firstly occurring much further above the  $K_{DP}$  peak and secondly has higher values than in FS1 and FS3. No collocated  $Z_{DR}$  signature exists, which can only mean that the  $Z_{DR}$  signature is completely masked by larger spherical particles. This fits to the theory of (Griffin et al., 2018): only very few and large D-type particles are present and produce high  $sZ_{DR_{max}}$  while  $K_{DP}$  is negligible due to their low number concentration. At T warmer than -15 °C,  $sZ_{DR_{max}}$  decreases while  $Z_e$  and  $K_{DP}$  increase. This signature might point to the early stage formation of D-type aggregates. If there was a layer of SLW at the top of the turbulent layer as mentioned by Grazioli et al. (2015), it is also possible that the dendrites from above become rimed, forming even graupel. The resulting velocity gradient in the PSD would favour collisions and hence collisional breakup. Resulting splinters would quickly grow into anisotropic ice crystals, enhance the number concentration and therefore  $K_{DP}$ . However, our data does not allow a definitive conclusion about the origin of this polarimetric signature.

Between around  $1200 \,\mathrm{m}$  and  $500 \,\mathrm{m}$  AGL corresponding to  $-12 \,\mathrm{and} - 7\,^{\circ}\mathrm{C}$ , FS4 is influenced by the turbulent layer (Fig. 4c and Fig. 8d) which is also evident in Fig. 9b) as strongly fluctuating MDV layer. In general, in FS4 the turbulent layer is more pronounced than during the previous fall streaks leading to a decrease in  $sZ_{DR_{max}}$  and  $Z_{DR}$  as well as an increase in SW.  $K_{DP}$  is rather homogeneous throughout FS4 inside the turbulent layer, horizontally and vertically. A decrease near the ground is only seen in the lowest  $200 \,\mathrm{m}$ . It seems likely that the constant  $K_{DP}$  is caused by increased ice collisional fragmentation that keep the number concentration of anisotropic particles high throughout the turbulent layer. VISSS images at the time of FS4 shows a wide mix of particles, many dendrite branches, aggregates in various stages of riming, and some graupel (see Fig. 6d). This confirms that ice collisional fragmentation and riming are likely major processes inside the turbulent layer. This is consistent with the findings of (e.g., Ramelli et al., 2021; Chellini and Kneifel, 2024). Similar to FS3, the ice collisional fragmentation in the observed turbulent layer might be the reason for the high  $N_{tot}$  of  $100 \,\mathrm{l}^{-1}$  during FS4.

## 3.3 Estimating particle size responsible for W-band $K_{DP}$ signatures using T-matrix simulations and CSU X-band $Z_e$ and $K_{DP}$ observations

In the previous section, we discussed the fingerprints of microphysical processes influencing polarimetric radar variables and hypothesized about the particles responsible for the observed  $K_{DP}$ . Here, we use T-matrix simulations and CSU X-band radar observations to constrain the sizes of particles contributing to  $K_{DP}$  signatures in the observed fall streaks. Instead of using LIMRAD94 data, the ratio of  $K_{DP}$  to  $Z_e$  at X-band is analyzed, as its  $Z_e$  is not affected by non-Rayleigh scattering for particle diameters smaller than  $10\,\mathrm{mm}$ .

Figure 10. T-matrix simulation results: Logarithm of the ratio between X-band  $K_{DP}$  and  $Z_e$  plotted for different aspect ratios and equivolumetric diameters (logarithmic x-axis). The black line is the median ratio per  $D_{equiv}$  over all aspect ratios (0.5-1.5). The minimum ratio per  $D_{equiv}$  can be seen for aspect ratios close to 1 and is plotted as blue dashed line. For aspect ratios <= 0.5 or >= 1.5 the ratio per  $D_{equiv}$  is higher (orange dashed line).

T-matrix simulations were performed to calculate the ratio of  $K_{DP}$  to  $Z_e$  at X-band for artificial spheroidal particles with varying aspect ratios. These calculations used  $Z_e$  in linear units (mm<sup>6</sup> m<sup>-3</sup>). Since both  $K_{DP}$  and  $Z_e$  are proportional to the concentration of particles, the ratio of these radar variables is therefore independent of the concentration. The ratio is thus mostly sensitive to size and shape of ice particles. The ratio decreases with an increase in particle diameter, because in snow,  $Z_e$  and  $K_{DP}$  are roughly proportional to the 4th and 1st power of the particle's size, respectively. Figure 10 shows the results of the T-matrix simulations, the logarithm of the ratio between  $K_{DP}$  and  $Z_e$  at X-band is plotted for different aspect ratios. Even though the  $K_{DP}$  and  $Z_e$  ratio depends on the aspect ratio of particles, it is suitable for roughly estimating size of particles producing  $K_{DP}$  signatures.

To assess the observed ratios of  $K_{DP}$  and  $Z_e$ , we analyzed CSU X-band data within the  $K_{DP}$  fall streaks (see Fig. 11). The CSU X-band  $K_{DP}$  was divided by  $\cos^2$  of the elevation angle to correct for the different elevation angles during the RHI scans (Ryzhkov and Zrnic, 2019). The  $Z_e$  and  $K_{DP}$  ratio was translated into  $D_{equiv}$  using T-matrix simulations shown in Fig. 10. Since the  $Z_e$  and  $K_{DP}$  ratio only slightly varies with the aspect ratio, we used the median value of the ratio for each  $D_{equiv}$ . The retrieved particle diameter ranges from 0.2 to 1.5 mm within the fall streaks. In the upper DGL, values tend to be less spread, ranging from 0.2 to 0.8 mm while they increase up to 1.5 mm towards the surface which might be linked to the

Figure 11. Time-height plot of LIMRAD94  $K_{DP}$  for the Dec 6, 2022 case study. CSU X-band data is superimposed on LIMRAD94  $K_{DP}$  when the latter is >= 1.5 ° km<sup>-1</sup>. Colored squares: Equivolumetric particle diameter retrieved from the logarithm of the ratio between X-band radar  $K_{DP}$  and  $Z_e$  for RHI scans. The analysed fall streaks as well as isotherms are indicated in grey.

increasing particle size through aggregation and/or depositional growth (also see Fig. 13). In a few pixels, the simulated sizes exceed 1.5 mm.

In FS1, simulated particle sizes below 1000 m exceed 0.8 mm more frequently compared to the other fall streaks. This aligns with the observed VISSS PSD in FS1, which stands out from the others by exhibiting a greater abundance of large aggregates and fewer small particles (see Fig. 7).

Considering that during the fall streaks, VISSS showed mean mass-weighted diameters ranging from 1.5 to 5 mm, we conclude that the contribution of large ice crystals to  $K_{DP}$  is small and a major part of the  $K_{DP}$  signatures is produced by ice particles smaller than 0.8 mm in diameter in the DGL and smaller than approximately 1.5 mm closer to the surface.

The ratio of  $K_{DP}$  and  $Z_e$  depends on the absolute calibration of CSU X-band. By comparing  $Z_e$  of CSU X-band with  $Z_e$  from LIMRAD94 (not shown), we found a low bias of CSU X-Band of about 4 dB. This roughly agrees with a low bias of 2.6 dB found with an absolute target calibration of the CSU X-Band performed pre-campaign. However, considering the several orders of magnitude range of the  $K_{DP}$  to  $Z_e$  ratios, even 4 dB offset would not significantly affect this analysis. In addition, we emphasize that the ratio alone is insufficient to estimate the exact size of particles when different populations of ice particles

coexist within the same volume. To illustrate this, consider two populations of particles, such as large aggregates and small ice particles. The large aggregates primarily determine  $Z_e$  due to their size, while the high concentration of small particles may dominate the  $K_{DP}$  values. Thus,  $Z_e$  and  $K_{DP}$  effectively characterize different particles within the volume. Nevertheless, the ratio can still be used to constrain the maximum possible size of particles dominating the  $K_{DP}$  values.

#### 3.4 DDA simulations of W-band $K_{DP}$ based on VISSS data

Based on the results obtained from the analysis of CSU X-band data, we expect that a major part of the  $K_{DP}$  signatures at W-band are produced by small ice particles. To further investigate the possible contribution of large particles, a more detailed analysis is required. The T-matrix method is not applicable to W-band measurements because large particles may produce resonance effects. In such cases, the spheroidal approximation does not always adequately represent the scattering properties (e.g. Schrom and Kumjian, 2018). To test the possible contribution of larger aggregates to our observed LIMRAD94  $K_{DP}$  signatures, we calculated  $K_{DP}$  and  $Z_e$  based on VISSS in situ observations using DDA simulations. The observations of LIMRAD94  $K_{DP}$  and  $Z_e$  are temporally averaged to 1 min and vertically averaged between 30 m (lowest range gate) and 400 m AGL. The ratio between observed and simulated  $K_{DP}$  and  $Z_e$  values at W-band are displayed in Fig.12 (a/b) (blue/black) solid line, respectively. For DDA simulations of  $K_{DP}$ , only particles >= 2.5 mm were used which we assume to be aggregates. We determined this size threshold based on images and the spectral complexity measurements from VISSS (not shown). The particles for the simulations were selected from the DDA database using a nearest neighbor regression, considering the nearest 5 particles and weighting them with their inverse distance to calculate the scattering properties expected for the in situ particle based on their maximum dimension. All of these particles were assumed to be dendrite aggregates, which fits the VISSS observations of particles larger than 2.5 mm.

Overall, the temporal evolution of the simulated  $K_{DP}$  and  $Z_e$  appears to match the observations quite well, except for the time during FS2 which is strongly influenced by blowing snow. During FS1, we see that the  $K_{DP}$  ratio is about 20 %, which is the value that Von Terzi et al. (2022) found as contribution of larger aggregates to W-band  $K_{DP}$  in their study. The simulated  $Z_e$  is about 80 to 100 % of the observed  $Z_e$ . This is about the range we would expect, given that the PSD during FS1 contains a comparably low number of particles < 2.5 mm. This also underlines that even when the larger aggregates determine  $Z_e$ , the smaller ice crystals still produce the majority of  $K_{DP}$  without contributing much to  $Z_e$ . This is consistent with our findings from the previous section 3.3.

During the cold front in the vicinity of FS2, there is only a very low number of larger aggregates and  $K_{DP}$  is mainly produced by small blowing snow particles as we explained in Sect. 3.2.3. Meanwhile, there are only few aggregates >= 2.5 mm (see blue line in Fig. 12c), hence their contribution to  $Z_e$  and  $K_{DP}$  is very small. During FS3, the number concentration of aggregates >= 2.5 mm is similar to FS1 while the number of particles < 2.5 mm is about two orders of magnitude higher than during FS1. This might explain why the simulated  $Z_e$  of particles >= 2.5 mm only explains 20-30 % of the observed  $Z_e$ . The observed  $K_{DP}$  is higher than during FS1 and the simulations suggest that larger aggregates contribute less than 10 % to the observed  $K_{DP}$ . Right before and after FS3 we see a higher number of larger aggregates which may contribute up to 15 % of the observed  $K_{DP}$ .

Figure 12. Panel a): Median LIMRAD94  $K_{DP}$  between 30 and 400 m AGL (red dashed line) with the 25-75th percentile (red shading) and ratio of DDA-based forward simulated  $K_{DP}$  (Sim.  $K_{DP}$ ) to observed LIMRAD94  $K_{DP}$  (blue line). For the blue line, only particles with diameter >= 2.5 mm were used, the blue shading shows the range of the simulations when just particles >= 3 mm (lower edge) or >= 1.6 mm (upper edge) are included. Panel b): Median LIMRAD94  $Z_e$  between 30 and 400 m AGL (green dashed line) with the 25-75th percentile (green shading) and ratio between DDA-based forward simulated  $Z_e$  (Sim.  $Z_e$ ) and observed LIMRAD94  $Z_e$  (black line). For the black line, only particles with diameter >= 2.5 mm were used, the grey shading shows the range of the simulations when just particles >= 3 mm (lower edge) or >= 1.6 mm (upper edge) are included. In panel c), VISSS particle number concentrations for particles with D >= 2.5 mm (blue) and D < 2.5 mm (orange) are shown. Magenta line marks the passage of a cold front while the black dotted lines mark the analyzed fall streaks.

During FS4, the number of particles >=  $2.5 \,\mathrm{mm}$  is highest compared to the other fall streaks, while the concentration of the smaller particles is slightly lower than during FS3. The contribution of bigger particles to  $K_{DP}$  is higher (around  $15\,\%$ ), while simulated  $Z_e$  of these particles contributes about 80 to  $90\,\%$  which seems reasonable.

In general, we conclude that a contribution to  $K_{DP}$  of 10-15 % can be attributed to aggregates with a maximum diameter >= 2.5 mm. When the PSD contains a comparably large number of aggregates, we even see contributions of up to 20 % to  $K_{DP}$  without overestimating  $Z_e$  in the simulations. In this analysis we completely omit the effects of riming. If riming enhances the density of aggregates,  $K_{DP}$  might be initially increased by an increase in density, provided that the aggregates maintain their anisotropic shape. The opposing effect of riming on  $K_{DP}$  was briefly discussed in 2.1.1.

#### 570 4 Summary and conclusions

We presented results obtained from in situ snowfall camera (VISSS) measurements and collocated polarimetric Doppler W-band (LIMRAD94) observations collected within the CORSIPP project embedded within the framework of the Atmospheric Radiation Measurement (ARM) Surface Atmosphere Integrated Field Laboratory SAIL campaign (Feldman et al., 2023). LIMRAD94  $K_{DP}$  and additional polarimetric variables along four high- $K_{DP}$  fall streaks during a cold-front passage related snowfall case study on December 6, 2022 were analyzed and linked to VISSS observations. To evaluate which particles contribute to the observed  $K_{DP}$  signatures, we further tried to constrain the particle sizes that contribute the most to the observed LIMRAD94  $K_{DP}$  using T-matrix simulations and collocated CSU X-band measurements. Lastly, we determined the amount of LIMRAD94  $K_{DP}$  likely produced by aggregates >= 2.5 mm using DDA simulations and VISSS in situ observations.

To conclude our study, we would like to highlight the following key findings:

- 1. By combining in situ and remote sensing measurements, we found that at W-band, high  $K_{DP}$  magnitudes are produced by a broad range of number concentrations  $(N_{tot})$ , ranging between 1 and  $100\,l^{-1}$  (see Sect. 3.1 and Fig. 2). Particle populations with  $N_{tot}$  below  $5\,l^{-1}$  often featured  $D_{32}$  above 5 mm, indicating that the PSD contains more large particles. We thus argue that when interpreting W-band  $K_{DP}$  signatures, it should be kept in mind that similar W-band  $K_{DP}$  values can be produced by strongly variable particle number concentrations.
- 2. Based on T-matrix simulations using a spheroidal approach and CSU X-band observations, we found that the main contributors to W-band  $K_{DP}$  signatures in the analyzed the fall streaks was likely by ice particles with a diameter below 0.8 mm in the DGL and below 1.5 mm closer to the surface (see Sect. 3.3 and Fig. 11). DDA simulations of  $K_{DP}$  based on VISSS PSDs during our analyzed the fall streaks show, that 10-15 %, in extreme cases up to 20 % of the  $K_{DP}$  observed close to the surface during the fall streaks can be attributed to aggregates >= 2.5 mm (see Sect. 3.4 and Fig. 12). This agrees with the results of Von Terzi et al. (2022). Revisiting the three possible explanations for high  $K_{DP}$  in particle populations with low  $N_{tot}$  and high  $D_{32}$  (Sect. 3.1), we conclude that small ice particles are the main contributors to the observed W-band  $K_{DP}$  and up to one-fifth of the observed  $K_{DP}$  can be attributed to large dendrite aggregates. When interpreting W-band  $K_{DP}$  signatures, it should be remembered that the contribution of aggregates to W-band  $K_{DP}$  is usually not negligible.

- 3. We identified blowing snow as another contributing source to increased  $K_{DP}$  values during a cold front passage. Specifically, lifted blowing snow likely caused W-band  $K_{DP}$  of up to 3 ° km<sup>-1</sup> (see Sect. 3.2.3).
- 4. Turbulence, more specifically spectral broadening and the increase in the width of the canting angle distribution, significantly reduces the values of  $Z_{DR}$  and  $K_{DP}$  significantly. The magnitude of  $sZ_{DR_{max}}$  seems to be more affected by turbulence than  $K_{DP}$ . Therefore, we conclude that it is essential to consider turbulence before interpreting any  $Z_{DR}$  or  $K_{DP}$  signatures.

- 5. Despite the dampening effect of turbulence mentioned above, we observed an increase in  $K_{DP}$  inside a strong turbulent layer. The turbulent layer was caused by wind shear along the interface between the blocked valley flow and the cross-barrier flow aloft (see Sect. 3.2.2). VISSS observations measured below the turbulent layers revealed number concentrations of over  $1001^{-1}$ . Due to high number of dendrite branches in VISSS observations, we suggest that this is caused by increased ice-collisional fragmentation that keeps the number concentration of anisotropic particles high throughout the turbulent layer and hence causes enhanced  $K_{DP}$ . This SIP mechanism of enhanced ice-collisional fragmentation in the presence of turbulence was also found in the studies of Ramelli et al. (2021) and Dedekind et al. (2023). Few graupel particles coexisting with unrimed ice observed by VISSS likely further increased the collision efficiency (see Sect. 3.2.3). Furthermore, we found an increase in  $sZ_{DR_{max}}$  and  $K_{DP}$  at the top of the turbulent layers during FS1 and FS2, as observed by (Hogan et al., 2002; Grazioli et al., 2015).
- 6.  $Z_{DR}$ ,  $sZ_{DR_{max}}$  and  $K_{DP}$  maxima of different magnitudes were well collocated in FS1 and FS3 and strongly displaced in FS4. This is likely linked to different concentrations of I-type and D-type particles as described in Griffin et al. (2018). When  $Z_{DR}$  signatures were masked by bigger and more spherical particles,  $sZ_{DR_{max}}$  proved to be a very useful additional variable. We therefore emphasize the advantage of using both, integrated and spectrally resolved  $Z_{DR}$ .

We have added a conceptual diagram (see Fig. 13) to provide an overview of the suspected dominant microphysical processes during the analyzed fall streaks, which we derived based on our observations.

During FS1, FS3 and FS4, we also hypothesized that early stage aggregates in the lower part of the DGL cloud have contributed to the observed W-band  $K_{DP}$  (as, for example, already discussed in Moisseev et al. (2015)). This speculation seems reasonable, as we did show that dendrite aggregates can contribute up to 20 % to W-band  $K_{DP}$ . However, in the observed case studies, the lower part of the DGL was about  $1000\,\mathrm{m}$  AGL. Thus, early stage aggregates were not observed at the ground by the VISSS. Early stage aggregates in the DGL might have a different shape and other polarimetric properties. A topic of further investigation of the CORSIPP dataset could thus be the analysis of polarimetric signatures of early stage aggregates for cases where the lower edge of the DGL was nearly at ground level.

We do not in particular focus on the effects of riming in our modeling studies. Riming however may even increase the contributions of aggregates to  $K_{DP}$ . In the early stages of riming, the density may be enhanced without an initial change in the particle shape (Erfani and Mitchell, 2017).  $K_{DP}$  might therefore initially increase by an increase in density, provided that the aggregates maintain their anisotropic shape. This influence of riming on aggregate contribution to  $K_{DP}$  may therefore be a

**Figure 13.** The suspected dominating microphysical processes during the analyzed fall streaks FS1-FS4) we derived based on our observations. The black arrows show for which temperature interval we assume a certain mechanism (marked with the respective letter) to be dominant. Snow particle images from VISSS (during the time of each fall streak) were used to visualize the possible particle types in each fall streak.

topic of further investigation. Naturally, there are caveats when linking surface in situ observations to radar measurements. The observational volume of LIMRAD94, even at close range, is several orders of magnitude larger than the one of VISSS. This can be partly compensated by temporal averaging and consideration of time shifts between the measurements, but generally requires us to assume some homogeneity in the precipitations.

The authors acknowledge that the measurement site is characterized by complex terrain and distinct synoptic forcing mechanisms. Consequently, the conclusions drawn are specific to this particular environmental and synoptic context and may not be generalizable to other settings.

In summary, this study highlights the complexity of interpreting W-band  $K_{DP}$  in snow. We show that polarimetric W-band radar observations combined with surface in situ observations of snow PSD by the VISSS and radar forward simulations can

give valuable insights into cloud and snowfall microphysical processes. Given the increasing use of polarimetric W-band radar observations and that even space-borne mission concepts have been proposed (Illingworth et al., 2018; Battaglia et al., 2022), the sensitivities of the polarimetric variables to microphysical properties should be further explored.

Table A1. Chirp table of LIMRAD94 used during CORSIPP.

| Attributes                                       | Chirp 1 | Chirp 2  | Chirp 3   | Chirp 4   |
|--------------------------------------------------|---------|----------|-----------|-----------|
| Integration time (s)                             | 0.654   | 0.950    | 1.198     | 1.198     |
| Range interval (m)                               | 40-493  | 493-2000 | 2000-6000 | 6000-9000 |
| Range vertical resolution (m)                    | 11.9    | 11.9     | 23.8      | 23.8      |
| Nyquist velocity (m s <sup>-1</sup> )            | 11.3    | 7.7      | 6.1       | 6.1       |
| Doppler velocity resolution (m s <sup>-1</sup> ) | 0.044   | 0.061    | 0.048     | 0.048     |
| Doppler velocity bins                            | 512     | 256      | 256       | 256       |

### Appendix A: Chirp tables of the polarimetric W-band radar

The chirp table used for the presented statistics and case studies is presented in Tab. A1.

**Figure B1.** Radar beam overlaps of LIMRAD94 and CSU X-band RHI scan (azimuth 329°) from sideview (panel a) and top view (panel b). Red lines represent the CSU X-band radar beam and black lines the LIMRAD94 radar beam. The size of the actual radar volume is not accounted for in the plot.

**Figure B2.** Vertical (panel a) and horizontal distance (panel b) of the closest radar beams from LIMRAD94 and CSU X-band during the RHI scans of CSU X-band plotted against the respective radar range. The colors indicate the respective elevation angle of CSU X-band.

Table B1. Locations and distances of instruments.

| Instrument | Latitude/Longitude         | Altitude AMSL [m] | Hor. dist. from LIMRAD94 [m] | Azimuth from LIMRAD94 [ $^{\circ}$ ] |
|------------|----------------------------|-------------------|------------------------------|--------------------------------------|
| VISSS      | 38.955912, -106.987834     | 2886              | 550                          | 151                                  |
| LIMRAD94   | 38.96031952, -106.99072266 | 2905              | -                            | -                                    |
| KAZR       | 38.956158, -106.987854     | 2886              | 525                          | 151                                  |
| CSU X-band | 38.89838791, -106.94324493 | 3147              | 8010                         | 149                                  |

#### Appendix B: Evaluation of antenna pointing and measurement comparability

The exact north alignment of LIMRAD94 was ensured by using a solar scan routine that is integrated into the LIMRAD94 software. The sun is a source of strong microwave radiation which can be detected by the passive 89 GHz channel of LIMRAD94. That way, the azimuth value at which the sun is visible in the measurements of LIMRAD94 is determined. The true azimuth of the sun is known from the GPS module onboard LIMRAD94. The offset from the true north is then simply the difference between the two azimuth values. During the first months of CORSIPP, LIMRAD94 measurements were performed at a constant elevation (CEL) of 40° and an azimuth of 151°. The field of view during the CEL measurements is shown in Fig. 1. The azimuth is chosen so that KAZR and VISSS are in the line of sight of LIMRAD94. The azimuth angle from LIMRAD94 towards CSU X-band is roughly 149° so there is a slight directional mismatch of both radar beams during the CEL measurements of LIMRAD94. The range height indicator (RHI) scans of CSU X-band towards an azimuth of 329° overlap best with the beam of LIMRAD94 (Fig. B1). However, the beams of both radars overlap enough to compare the measurements. In order to make the RHI scans of CSU X-band and the CEL measurements of LIMRAD94 comparable, the closest range gates were selected for each time step. With increasing altitude, hence the elevation angle of CSU X-band, the horizontal distance of the closest range gates increases. The maximum horizontal distance is 120 m, the vertical distance does not exceed 8 m (Fig. B2). Coordinates and relative locations of all instruments can be found in Tab. B1.

Author contributions. AK collected and processed the LIMRAD94 data from CORSIPP, analyzed and plotted the data and wrote the manuscript with contributions from all coauthors. LT and DO provided the DDA simulations, LT wrote the section 2.4.2. AM provided the T-matrix simulations and wrote the Sect. 2.4.1. VE created Fig. 6 and wrote the caption. TV processed the EDR data. MM processed the VISSS data. HKL and MM acquired funding and guided the research project. AK, AM, MM, VE, AR, PB, LT, DO, HKL participated in joint meetings to discuss the interpretation of polarimetric variables observed during the presented case study. All authors reviewed and edited the manuscript. The AI tool Writefull was used to correct spelling and typesetting.

Competing interests. The authors declare that they have no competing interests.

675

680

Code and data availability. SAIL data were obtained from the Atmospheric Radiation Measurement (ARM) user facility, a U.S. Department of Energy (DOE) Office of Science user facility managed by the Biological and Environmental Research Program.: LIMRAD94 (https://doi.org/10.5439/2229846, last access: December 5, 2024), VISSS (https://doi.org/10.5439/2278627, last access: December 5, 2024), the meteorological in situ 415 data of AMF2 (https://doi.org/10.5439/1786358, last access: December 5, 2024), the microwave radiometer retrieval products (https://doi.org/10.5439/1027369, last access: 5 Dec 2024), the ARM KAZR (https://doi.org/10.5439/1498936, last access: 5 Dec 2024), the CSU X-band (https://doi.org/10.5439/1888379, last access: 5 Dec 2024).

Acknowledgements. This research is funded by the Deutsche Forschungsgemeinschaft (DFG, German Research Foundation) - Project number: 408008112 (Characterization of orography-influenced riming and secondary ice production and their effects on precipitation rates using radar polarimetry and Doppler spectra - CORSIPP) within the Priority Program SPP 2115 "Polarimetric Radar Observations meet Atmospheric Modelling (PROM) – Fusion of Radar Polarimetry and Numerical Atmospheric Modelling Towards an Improved Understanding of Cloud and Precipitation Processes". Contributions by L. von Terzi have been supported by the DFG Priority Program SPP2115 "Fusion of Radar Polarimetry and Numerical Atmospheric Modelling Towards an Improved Understanding of Cloud and Precipitation Processes" (PROM) under grant PROM-FRAGILE (project number 492234709). This research was supported by the Atmospheric Radiation Measurement (ARM) user facility, a U.S. Department of Energy (DOE) Office of Science user facility managed by the Biological and Environmental Research Program. This work was supported in part by the European Space Agency under the activity WInd VElocity Radar Nephoscope (WIVERN) Phase A Science and Requirements Consolidation Study, ESA Contract Number 4000144120/24/NL/IB/ab. We would like to thank the Rocky Mountain Biological Laboratory staff as well as ARM technicians for logistical and on-site technical support. We would like to thank Dr. Stefan Kneifel for his valuable feedback during our meetings discussing the paper material. We would also like to thank the PI of the CSU X-band radar Dr. V Chandrasekar and his team for providing access to the CSU X-band data set via the ARM data portal.

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
