# Peer review of "Investigating KDP signatures inside and below the dendritic growth layer with W-band Doppler Radar and in situ snowfall camera"

_EGUsphere, 2025_

## Author Comment (AC1)

**Author's response to:**
**RC#1**
**https://doi.org/10.5194/egusphere-2025-734-RC1**

Anton Kötsche[1], Alexander Myagkov[2], Leonie von Terzi[4], Maximilian Maahn[1], Veronika Ettrichrätz[1], Teresa Vogl[1], Alexander Ryzhkov[5,6], Petar Bukovcic[5,6], Davide Ori[3], and Heike Kalesse-Los[1]

[1]Leipzig Institute for Meteorology (LIM), Leipzig University, Leipzig, Germany
[2]Radiometer Physics GmbH, Meckenheim, Germany
[3]Institute of Geophysics and Meteorology, University of Cologne, Cologne, Germany
[4]Meteorological Institute, Ludwig-Maximilians-Universität in Munich, Munich, Germany
[5]NOAA National Severe Storms Laboratory, Norman, Oklahoma, USA
[6]Cooperative Institute for Severe and High-Impact Weather Research and Operations, University of Oklahoma, Norman, Oklahoma, USA

**Correspondence:** anton.koetsche@uni-leipzig.de

Dear Reviewer,

Thank you for carefully reading the manuscript and pointing out several issues where the description needs to be refined for a better understanding. The requested clarifications and references to ambiguities contribute to the improvement of the manuscript.

To separate the Reviewer's comments and the author's response, we printed the comments in black and the response in blue. Excerpts of the manuscript with marked changes are pinned directly to the appropriate responses, with the indicated text location (e.g., line number) referring to the manuscript in the preprint.

Sincerely, on behalf of all authors

Anton Kötsche

**Summary of main changes of the manuscript:**

– Changed color and labels in Fig. 8.

– Added conceptual diagram showing the microphysical mechanisms (as new Fig. 13) in conclusion.

– Figure with T-matrix simulations (Fig. 10) was modified to show the connections between $K_{DP}/Z_e$ ratio, $D_{equiv}$ and the aspect ratio more easily.

– $D_{equiv}$ of particles were directly used as colors (instead of $K_{DP}/Z_e$ ratio) in the Fig. 11 and the text in Section 3.3 was adapted.

– Added chapter **Rationale for the applicability of scattering models** as new section 2.4.3 to substantiate our decision of using T-Matrix and DDA.

– The slope parameter $\Lambda$ from the VISSS PSDs was added to Fig. 4 and added to argumentation.

[Figure]

**Figure 1.** The suspected dominating microphysical processes during the analyzed fall streaks FS1-FS4) we derived based on our observations. The black arrows show for which temperature interval we assume a certain mechanism (marked with the respective letter) to be dominant. Snow particle images from VISSS (during the time of each fall streak) were used to visualize the possible particle types in each fall streak.

**Response to RC#1:**

**Major comment #1: Schematic**

*I think a conceptual diagram showing the microphysical mechanisms occurring in each fall streak observed should be in the paper, perhaps in the conclusions section. This would make it easier for the reader to envision the complex microphysics of these fall streaks.*

We added Fig. 1 to illustrate the key mechanisms during the fall streaks. We also added the following short paragraph in the conclusion part:

> **L615f.:**
>
> A conceptual diagram was added (see Fig. 13) to give an overview over the suspected dominating micro-physical processes during the analyzed fall streaks we derived based on our observations.

[Figure]

**Figure 2.** T-matrix simulation results: Logarithm of the ratio between X-band $K_{DP}$ and $Z_e$ plotted for different aspect ratios and equivolumetric diameters (logarithmic x-axis). The black line is the median ratio per $D_{equiv}$ over all aspect ratios (0.5-1.5). The minimum ratio per $D_{equiv}$ can be seen for aspect ratios close to 1 and is plotted as blue dashed line. For aspect ratios <= 0.5 or >= 1.5 the ratio per $D_{equiv}$ is higher (orange dashed line).

**Major comment #2: KDP Modeling using T-Matrix**

*In Section 3.3, the authors use T-matrix simulations to estimate dual polarimetric radar moments from particle size distribution of aggregates at X-band, but then proceed to use DDA for their W-band simulated moment calculations. This leaves the reader wondering why DDA was not used for both the X-band and W-band scattering simulations, since DDA should be much better able to characterize the scattering of irregular ice crystals than T-Matrix based techniques. Why were T-Matrix calculations used for simulating the X-band KDP/Ze? I think it would be more consistent to use DDA for both wavelengths. Either that, or the authors should elucidate as to why they chose T-Matrix for their X-band simulated KDP/Ze values. Perhaps the authors wanted to easily determine how aspect ratio can determine the simulated KDP/Ze. If so, that should be better stated here. I also do not think Figure 10 is needed, as the KDP/Ze ratio seems to only significantly vary with size, with a few hundred microns uncertainty in the size due to oblateness. These estimated ranges of particle sizes can simply be quoted in the text.*

We would like to clarify that the primary objective of using scattering models in our manuscript is not to develop a retrieval method, but rather to estimate the types of particles that frequently produce observed KDP signatures. Since the T-matrix approximation uses spheroids, a clear relation between the KDP/Ze ratio and the size and aspect ratio of the spheroid for a

[Figure]

**Figure 3.** As Figure 10 in the original manuscript, but produced using DDA calculations of 2627 ice crystals.

fixed apparent ice density is possible. As for DDA we use more detailed shapes, fixing the effective density is not possible and such a clear relationship can not be found. Further, we do not have precise information about the size or shape of the observed particles. It is therefore not clear which shape from the DDA database to use, especially critical could be the decision between aggregates and ice crystals, since the difference in density between ice crystals and aggregates is large, and therefore potentially also the KDP/Ze ratio. Nevertheless, as a large DDA-database consisting of 2,627 ice crystals including varying dendritic shapes, plates and columns is available to us, we chose to calculate the same plot with DDA, assuming ice crystals as a particle type (see Figure 3). While there is some variability, especially for the oblate like particles (in this case dendrites and plates), owing to the large variability in dendrite shapes, and therefore variability in density, the overall behaviour is similar as when computing the plot with the Tmatrix. As dendrites and columns typically have aspect ratios smaller than 0.01 and 10 respectively, we also chose to "zoom out" of the original axis which were constrained between 0.5 and 1.5 (see Figure 4). In this plot it is evident that there is also a slight gradient of KDP/Ze for decreasing (oblate) / increasing (prolate) aspect ratios. This gradient is however much smaller compared to the gradient observed with increasing size. The comparison with DDA showed that in this case the T-matrix can be used to calculate the approximate relationship between the size and KDP/Ze ratio. While the T-matrix might be able to calculate the scattering properties of ice crystals with a large effective density accurately,

[Figure]

**Figure 4.** As Figure 3, but showing the full range of aspect ratios available in the DDA Database.

it can not be used to calculate the scattering properties of ice aggregates. The largest problem is assuming an aggregate to be a soft spheroid, since the density in the soft spheroid has to be assumed to be really small to fit the mass and maximum dimension of the aggregate. This causes especially the polarimetric signatures of aggregates to be underestimated largely by the T-matrix. Further, when the size of the particle and the wavelength of the radar become comparable, the distribution of the mass inside the particle is important for its scattering properties (Ori et al., 2021, and references therein). For the simulations in section 3.4, it was clear from the VISSS observations that aggregates are the predominant particle type. Therefore, for estimating KDP and Ze for the particles observed with the VISSS, we are choosing DDA calculations. We decided to directly use the equivolumetric diameter of particles as colors in the Fig. 11 (see Fig. 5). We altered Fig. 10 so that the KDP/Ze ratio used for Fig. 11 and also the variation with aspect ratio can be determined more easily (see Fig. 2). We added the following section to clarify our decision for the scattering models in the paper.

**219f.:**
* * *
**Rationale for the applicability of scattering models** The primary aim of employing scattering models in this study is to identify the types of ice particles that contribute to the observed $K_{DP}$ signatures, rather than to establish a retrieval method. Since the T-matrix approximation uses spheroids, a clear relation between the $K_{DP}/Z_e$ ratio and the size and aspect ratio of the spheroid for a fixed apparent ice density is possible. While this approach is simplified, it yields sufficiently accurate results for preliminary conclusions, particularly when the particles are smaller than the radar wavelength, where the accurate mass distribution inside of the simulated particle is not as important as when particle size and wavelength become comparable. As for DDA we use more detailed shapes, fixing the effective density is not possible and such a clear relationship can not be found. Further, we do not have precise information about the size or shape of the observed particles. We therefore applied the spheroidal approximation and T-matrix model at X-band, the lowest frequency available in our campaign with a collocated measurement volume, to assess the sensitivity of $K_{DP}$ and $Z_e$ to particle size and aspect ratio (Sect. 3.3). This analysis also aids in determining whether $K_{DP}$ signatures are predominantly generated by larger particles. Conversely, the T-matrix method produces larger errors at W-band frequencies, where the shorter wavelength means non-Rayleigh scattering effects and the accurate description of the mass distribution in the particle become relevant (e.g. Ori et al., 2021, and references therein). This is especially true for particles with a low density, such as aggregates. Describing an aggregate with a soft spheroid is problematic, as the effective density in the spheroid needs to be small to accurately match the size and mass of the aggregate. This causes the (polarimetric) scattering properties of aggregates to be underestimated by soft spheroid methods such as the T-matrix (e.g. Ori et al., 2021). Consequently, to estimate $K_{DP}$ and $Z_e$ at W-band for the particles detected with the VISSS, particularly concentrating on dendrite aggregates, we opt to use DDA computations (Sect. 3.4).

**Major comment #3: KDP/Ze ratio**

*Lines 484-490: There are log10 (KDP/Ze) values down to -4 in FS1. Therefore, statements here about log10 (KDP/Ze) being greater than -2.5 do not look to apply here. In addition, it looks like this ratio increases with time as we see the PSD from the VISSS record higher concentrations of small particles, though this would require plotting timeseries of log10(KDP/Ze) for a few given levels to verify. These lower KDP/Ze, especially in FS1, would be consistent with the larger aggregates/graupel below the -10 C level as suggested from the vertical profiles from LIMRAD94 in Figure 8. Since the VISSS-observed PSD in FS1 is radically different from the other 3, I think these points are worth factoring into your discussion of Figure 11.*

We thank the reviewer for the helpful suggestions. The text was modified as follows:

[Figure]

**Figure 5.** Time-height plot of LIMRAD94 $K_{DP}$ for the Dec 6, 2022 case study. CSU X-band data is superimposed on LIMRAD94 $K_{DP}$ when the latter is >= $1.5\,^{\circ}\,\mathrm{km}^{-1}$. Colored squares: Equivolumetric particle diameter retrieved from the logarithm of the ratio between X-band radar $K_{DP}$ and $Z_e$ for RHI scans. The analysed fall streaks as well as isotherms are indicated in grey.
* * *
**L508f.:**
* * *
To assess the observed ratios of KDP and Ze, we analyzed CSU X-band data within the KDP fall streaks (see Fig. 11). The CSU X-band KDP was divided by cos2 of the elevation angle to correct for the different elevation angles during the RHI scans (Ryzhkov and Zrnic, 2019). The  Ze and KDP ratio was translated into $D_{equiv}$ using T-matrix simulations shown in Fig. 10. Since the Ze and KDP ratio only slightly varies with the aspect ratio, we used the median value of the ratio for each $D_{equiv}$. The retrieved particle diameter ranges from 0.2 to 1.5 mm within the fall streaks. In the upper DGL, values tend to be less spread, ranging from  0.2 to 0.8 mm while they increase up to 1.5 mm towards the surface which might be linked to the increasing particle size through aggregation and/or depositional growth (also see Fig. 13).  In a few pixels, the simulated sizes exceed 1.5 mm. In FS1, simulated particle sizes below 1000 m exceed 0.8  mm more frequently compared to the other FS. This aligns with the observed VISSS PSD in FS1, which stands out from the other fall streaks by exhibiting a greater abundance of large aggregates and fewer small particles (see Fig. 7).Considering that during the fall streaks, VISSS showed mean mass-weighted diameters ranging from 1.5 to 5 mm, we conclude that the contribution of large ice crystals to KDP is small and a major part of the KDP signatures is produced by ice particles smaller than 0.8 mm in diameter in the DGL and smaller than approximately  1.5 mm closer to the surface.

**Minor comments**

- *Line 8: It would be nice to be more quantitative about what magnitudes of Kdp we are talking about here.*

- We agree: We have rephrased this paragraph as:

  > **L8f.**:
  >
  > We found that at W-band,  $K_{DP}$  $> 2\,^{\circ}\,\mathrm{km}^{-1}$ can result from a broad range of particle number concentrations, between 1 and $100\,\mathrm{l}^{-1}$.

- *Line 492: I know that an absolute target calibration of the CSU X-Band was performed pre-campaign. That showed that the CSU X-band had a low bias of 2.6 dB, so it would be worth stating that here. It's also not clear here if the stated offset is a low offset or a high offset.*

- We thank you for this additional information. We have rephrased this paragraph as:

  > **525f.**:
  >
  > The ratio of $K_{DP}$ and $Z_e$ depends on the absolute calibration of CSU X-band. By comparing $Z_e$ of CSU X-band with $Z_e$ from LIMRAD94 (not shown), we found a low bias of CSU X-Band of about 4 dB. This roughly agrees with a low bias of 2.6 dB found with an absolute target calibration of the CSU X-Band performed pre-campaign.

- *Figure 8: The purple color for spectral width is hard to distinguish from the ZDR and sZDR curves. I would suggest changing the color to Orange. I also would remove the figure legends from sub-panels b, c, and d to increase readability.*

- We agree, the plot has been updated according to your suggestion.

**Technical Comments**

- *Line 16: The first sentence in Section 1 is a run-on sentence*

- Replaced first comma by semicolon.

- *Line 126: Missing parentheses around citation.*

- Added parentheses

- *Line 167: Extra "," after "40."*

- Fixed

**References**

Ori, D., von Terzi, L., Karrer, M., and Kneifel, S.: snowScatt 1.0: consistent model of microphysical and scattering properties of rimed and unrimed snowflakes based on the self-similar Rayleigh–Gans approximation, Geoscientific Model Development, 14, 1511–1531, https://doi.org/10.5194/gmd-14-1511-2021, publisher: Copernicus GmbH, 2021.

---

## Author Comment (AC2)

**Author's response to:**
**RC#2**
**https://doi.org/10.5194/egusphere-2025-734-RC2**

Anton Kötsche[1], Alexander Myagkov[2], Leonie von Terzi[4], Maximilian Maahn[1], Veronika Ettrichrätz[1], Teresa Vogl[1], Alexander Ryzhkov[5,6], Petar Bukovcic[5,6], Davide Ori[3], and Heike Kalesse-Los[1]

[1]Leipzig Institute for Meteorology (LIM), Leipzig University, Leipzig, Germany
[2]Radiometer Physics GmbH, Meckenheim, Germany
[3]Institute of Geophysics and Meteorology, University of Cologne, Cologne, Germany
[4]Meteorological Institute, Ludwig-Maximilians-Universität in Munich, Munich, Germany
[5]NOAA National Severe Storms Laboratory, Norman, Oklahoma, USA
[6]Cooperative Institute for Severe and High-Impact Weather Research and Operations, University of Oklahoma, Norman, Oklahoma, USA

**Correspondence:** anton.koetsche@uni-leipzig.de

Dear Reviewer,

Thank you for carefully reading the manuscript and pointing out several issues where the description needs to be refined for a better understanding. The requested clarifications and references to ambiguities contribute to the improvement of the manuscript.

To separate the Reviewer's comments and the author's response, we printed the comments in black and the response in blue. Excerpts of the manuscript with marked changes are pinned directly to the appropriate responses, with the indicated text location (e.g., line number) referring to the manuscript in the preprint.

Sincerely, on behalf of all authors

Anton Kötsche

**Summary of main changes of the manuscript:**

- Changed color and labels in Fig. 8.

- Added conceptual diagram showing the microphysical mechanisms (as new Fig. 13) in conclusion.

- Figure with T-matrix simulations (Fig. 10) was modified to show the connections between $K_{DP}/Z_e$ ratio, $D_{equiv}$ and the aspect ratio more easily.

- $D_{equiv}$ of particles were directly used as colors (instead of $K_{DP}/Z_e$ ratio) in the Fig. 11 and the text in Section 3.3 was adapted.

- Added chapter **Rationale for the applicability of scattering models** as new section 2.4.3 to substantiate our decision of using T-Matrix and DDA.

- The slope parameter $\Lambda$ from the VISSS PSDs was added to Fig. 4 and added to argumentation.

[Figure]

**Figure 1.** As Figure 10 in the original manuscript, but produced using DDA calculations of 2627 ice crystals.

**Response to RC#2:**

**Major comment #1: KDP Modeling using T-Matrix**

*Although the CSU X-band radar is less susceptible to non-Rayleigh scattering, why was the T-matrix method used over DDA to remain consistent with the W-band polarization analysis (Sec. 3.3)? At a minimum, the authors should more clearly defend this decision (e.g., computational cost, etc.).* We would like to clarify that the primary objective of using scattering models in our manuscript is not to develop a retrieval method, but rather to estimate the types of particles that frequently produce observed KDP signatures. Since the Tmatrix approximation uses spheroids, a clear relation between the KDP/Ze ratio and the size and aspect ratio of the spheroid for a fixed apparent ice density is possible. As for DDA we use more detailed shapes, fixing the effective density is not possible and such a clear relationship can not be found. Further, we do not have precise information about the size or shape of the observed particles. It is therefore not clear which shape from the DDA database to use, especially critical could be the decision between aggregates and ice crystals, since the difference in density between ice crystals and aggregates is large, and therefore potentially also the KDP/Ze ratio. Nevertheless, as a large DDA-database consisting of 2,627 ice crystals including varying dendritic shapes, plates and columns is available to us, we chose to calculate the same plot

[Figure]

**Figure 2.** As Figure 1, but showing the full range of aspect ratios available in the DDA Database.

with DDA, assuming ice crystals as a particle type (see Figure 1). While there is some variability, especially for the oblate like particles (in this case dendrites and plates), owing to the large variability in dendrite shapes, and therefore variability in density, the overall behaviour is similar as when computing the plot with the Tmatrix. As dendrites and columns typically have aspect ratios smaller than 0.01 and 10 respectively, we also chose to "zoom out" of the original axis which were constrained between 0.5 and 1.5 (see Figure 2). In this plot it is evident that there is also a slight gradient of KDP/Ze for decreasing (oblate) / increasing (prolate) aspect ratios. This gradient is however much smaller compared to the gradient observed with increasing size. The comparison with DDA showed that in this case the Tmatrix can be used to calculate the approximate relationship between the size and KDP/Ze ratio. While the Tmatrix might be able to calculate the scattering properties of ice crystals with a large effective density accurately, it can not be used to calculate the scattering properties of ice aggregates. The largest problem is assuming an aggregate to be a soft spheroid, since the density in the soft spheroid has to be assumed to be really small to fit the mass and maximum dimension of the aggregate. This causes especially the polarimetric signatures of aggregates to be underestimated largely by the Tmatrix. Further, when the size of the particle and the wavelength of the radar become comparable, the distribution of the mass inside the particle is important for its scattering properties (Ori et al., 2021, and references therein). For the simulations in section 3.4, it was clear from the VISSS observations that aggregates are the

[Figure]

**Figure 3.** T-matrix simulation results: Logarithm of the ratio between X-band $K_{DP}$ and $Z_e$ plotted for different aspect ratios and equivolumetric diameters (logarithmic x-axis). The black line is the median ratio per $D_{equiv}$ over all aspect ratios (0.5-1.5). The minimum ratio per $D_{equiv}$ can be seen for aspect ratios close to 1 and is plotted as blue dashed line. For aspect ratios <= 0.5 or >= 1.5 the ratio per $D_{equiv}$ is higher (orange dashed line).

predominant particle type. Therefore, for estimating KDP and Ze for the particles observed with the VISSS, we are choosing DDA calculations. We decided to directly use the equivolumetric diameter of particles as colors in the Fig. 11 (see Fig. 4). We altered Fig. 10 so that the KDP/Ze ratio used for Fig. 11 and also the variation with aspect ratio can be determined more easily (see Fig. 3). We added the following section to clarify our decision for the scattering models in the paper.

> **219f.:**
>
> ---
>
> **Rationale for the applicability of scattering models** The primary aim of employing scattering models in this study is to identify the types of ice particles that contribute to the observed $K_{DP}$ signatures, rather than to establish a retrieval method. Since the T-matrix approximation uses spheroids, a clear relation between the $K_{DP}/Z_e$ ratio and the size and aspect ratio of the spheroid for a fixed apparent ice density is possible. While this approach is simplified, it yields sufficiently accurate results for preliminary conclusions, particularly when the particles are smaller than the radar wavelength, where the accurate mass distribution inside of the simulated particle is not as important as when particle size and wavelength become comparable. As for DDA we use more detailed shapes, fixing the effective density is not possible and such a clear relationship can not be found. Further, we do not have precise information about the size or shape of the observed particles. We therefore applied the spheroidal approximation and T-matrix model at X-band, the lowest frequency available in our campaign with a collocated measurement volume, to assess the sensitivity of $K_{DP}$ and $Z_e$ to particle size and aspect ratio (Sect. 3.3). This analysis also aids in determining whether $K_{DP}$ signatures are predominantly generated by larger particles. Conversely, the T-matrix method produces larger errors at W-band frequencies, where the shorter wavelength means non-Rayleigh scattering effects and the accurate description of the mass distribution in the particle become relevant (e.g. Ori et al., 2021, and references therein). This is especially true for particles with a low density, such as aggregates. Describing an aggregate with a soft spheroid is problematic, as the effective density in the spheroid needs to be small to accurately match the size and mass of the aggregate. This causes the (polarimetric) scattering properties of aggregates to be underestimated by soft spheroid methods such as the T-matrix (e.g. Ori et al., 2021). Consequently, to estimate $K_{DP}$ and $Z_e$ at W-band for the particles detected with the VISSS, particularly concentrating on dendrite aggregates, we opt to use DDA computations (Sect. 3.4).

**Major comment #2: Particle habits**

*Aside from the fallstreak analysis from one case (Fig. 6), there is little analysis on the observed particle types/habits observed from the VISS. Have the authors considered identifying these habits statistically to confirm the presence of e.g. dendrites versus much more commonly-occurring irregular crystals? If identifying these habits is too time consuming or beyond the scope of this study, perhaps the VISSS complexity parameter can elucidate the connection of more complex shapes, such as dendrites, to the polarimetric observations in the environments conducive to such particle habits. It is important to provide evidence of the particle types that are related to the microphysical processes discussed (e.g., dendrites in the DGZ, needles with SIP).*

Analyzing particle types from VISSS images by an algorithm is something we are currently working on in our group, a separate paper on this is planned for later this year. Obtaining reliable results with this method will however require more time and would be beyond the scope of this study. We tried to provide evidence of the particle types in Fig. 6 where we show exemplary images of snow particles throughout the PSD. Please see Fig. 5 for a view of the complexity spectrum during the

[Figure]

**Figure 4.** Time-height plot of LIMRAD94 $K_{DP}$ for the Dec 6, 2022 case study. CSU X-band data is superimposed on LIMRAD94 $K_{DP}$ when the latter is >= 1.5 $^\circ$ km$^{-1}$. Colored squares: Equivolumetric particle diameter retrieved from the logarithm of the ratio between X-band radar $K_{DP}$ and $Z_e$ for RHI scans. The analysed fall streaks as well as isotherms are indicated in grey.

case study presented in the paper. During FS1, particles larger than 1 mm are more rimed (lower complexity) which is also visible in Fig. 6 in the paper and which we addressed in the FS1 description. It's also evident that particles larger than 2.5 mm are aggregates, this size threshold was used in section 3.4 of the paper for the DDA calculations. Otherwise its hard to make out certain particle types just based on complexity, therefore we choose to use real VISSS images instead to for example show needles or dendrite fragments. In our opinion this is sufficient to give evidence of the present particle types.

**Major comment #3: Other VISSS parameters**

*Deriving other PSD parameters from the VISSS, such as the slope (Λ) or shape (μ) parameter may further strengthen the relationships/comparisons between the microphysics and remotely-sensed regions of cloud. Statements such as "It is likely that aggregation rapidly depletes the number of small particles" (L376) read as speculative when these parameters can be used to strengthen the arguments made.*

We added the slope of the PSD as a variable in Fig. 4 (see Fig. 6 in this answer). The following modifications/additions were made to the text:

[Figure]

**Figure 5.** VISSS derived complexity spectrum during the case study presented in the paper.
* * *
**175f.:**

Following (Maahn et al., 2024), the complexity $c$ is derived from the ratio of the particle perimeter $p$ to the perimeter of a circle with the same area $A$ :

$$c = \frac{p}{2\sqrt{\pi A}}. \tag{1}$$

The slope parameter of the PSD ($\Lambda$) is derived using the definition of Maahn et al. (2015):

$$\Lambda = N_0/N_{tot} \tag{2}$$

where $N_0$ is the intercept parameter of the exponantial PSD.
* * *
**310f.:**

The nature of the precipitation changed during the cold front passage; from low Ntot, low $\Lambda$ and high D32 precipitation dominated by aggregates and comparably low number of small particles before the cold front to a high Ntot, high $\Lambda$ and low D32 precipitation with very mixed particle types (see Fig. 4e).
* * *
**354f.:**

A low $\Lambda$ (Fig. 4e) of around $0.5\,\mathrm{mm}^{-1}$ further suggests a PSD with a reduced number of smaller particles, which is also visible in Fig. 7)

> **401f.:**
>
>  Aggregation likely causes a rapid depletion of small particles  near the surface, thereby reducing Ntot. The low Λ observed in Fig. 4 provides additional evidence that small particles are depleted in the PSD.

**Major comment #4: Schematic**

*The summary/conclusions section may benefit from a conceptual diagram that visualizes the complexities associated with the microphysics and their relation to KDP, etc. as alluded to in the abstract and introduction*

We added Fig. 7 to illustrate the key mechanisms during the fall streaks. We also added the following short paragraph in the conclusion part:

> **L615f.:**
>
> We have added a conceptual diagram (see Fig. 13) to provide an overview of the suspected dominant microphysical processes during the analyzed fall streaks, which we derived based on our observations

**Minor comments**

- *Fig. 7: Please confirm whether what's being plotted is a concentration, as the y-axis units suggests, or if it's normalized by the bin width which would require the units to be corrected.*

- Thanks for noticing, the unit was not correct, we fixed it.

- *Because this study uses data collected near mountainous terrain, it should be stated somewhere that these conclusions represent a particular environment/synoptic setup and may not be valid for all environments (e.g., different forcing mechanisms, presence of supercooled liquid, etc.).*

- We agree and have added the following paragraph in the conclusion:

  > **632f.:**
  >
  > The authors acknowledge that the measurement site is characterized by complex terrain and distinct synoptic forcing mechanisms. Consequently, the conclusions drawn are specific to this particular environmental and synoptic context and may not be generalizable to other settings.

**Technical Comments**

Thank you for the technical comments, we incorporated all of them as suggested.

- *L55: You can remove "among others" as the "e.g." implies this*

[Figure]

**Figure 6.** LIMRAD94 time-height plots during the case study on December 6, 2022. Analyzed fall streaks shown as black lines with labels. In every panel except d), data is only colored where LIMRAD94 KPD $> 2\,^\circ\,\mathrm{km}^{-1}$. The remainder of the $Z_e$ and $sZ_{DR_{max}}$ / $Z_{DR}$ observations where $K_{DP}$ was below this threshold are displayed in black and white. Grey lines are isotherms. a) $Z_e$. b) $Z_{DR}$. c) $sZ_{DR_{max}}$ with areas masked where $EDR$ exceeds the turbulence threshold of $0.001\,\mathrm{m}^2\,\mathrm{s}^{-3}$. d) $K_{DP}$. e) Minutely VISSS $N_{tot}$ on the y-axis and $\Lambda$ (black line) plotted against time on the x-axis. Colors are the respective $D_{32}$ values.

[Figure]

**Figure 7.** The suspected dominating microphysical processes during the analyzed fall streaks FS1-FS4) we derived based on our observations. The black arrows show for which temperature interval we assume a certain mechanism (marked with the respective letter) to be dominant. Snow particle images from VISSS (during the time of each fall streak) were used to visualize the possible particle types in each fall streak.

– *L75: Remove "e.g."*

– *L84: I recommend capitalizing the words comprising of the CORSIPP acronym*

– *L103: "overview over" -> "overview of"*

– *L113: Add a comma after "respectively"*

– *L150: Parentheses are only needed around the units*

– *L150: "radar reflectivity" -> "equivalent radar reflectivity"*

– *L250: Add a space between 500 m*

– *L321: "Chapter" -> "Section"*

**References**

Maahn, M., Löhnert, U., Kollias, P., Jackson, R. C., and McFarquhar, G. M.: Developing and Evaluating Ice Cloud Parameterizations for Forward Modeling of Radar Moments Using in situ Aircraft Observations, https://doi.org/10.1175/JTECH-D-14-00112.1, section: Journal of Atmospheric and Oceanic Technology, 2015.

Maahn, M., Moisseev, D., Steinke, I., Maherndl, N., and Shupe, M. D.: Introducing the Video In Situ Snowfall Sensor (VISSS), Atmospheric Measurement Techniques, 17, 899–919, https://doi.org/10.5194/amt-17-899-2024, publisher: Copernicus GmbH, 2024.

Ori, D., von Terzi, L., Karrer, M., and Kneifel, S.: snowScatt 1.0: consistent model of microphysical and scattering properties of rimed and unrimed snowflakes based on the self-similar Rayleigh–Gans approximation, Geoscientific Model Development, 14, 1511–1531, https://doi.org/10.5194/gmd-14-1511-2021, publisher: Copernicus GmbH, 2021.